



# How wrong are climate field reconstruction techniques in reconstructing a climate with long-range memory?

Tine Nilsen [1], Johannes P. Werner [2], and Dmitry V. Divine [3, 1]

[1]Department of Mathematics and Statistics, University of Tromsø The Arctic University of Norway, Tromsø, Norway
[2]Bjerknes Centre for Climate Research and Department for Earth Science, University of Bergen, Bergen, Norway
[3]Norwegian Polar Institute, Tromsø, Norway

*Correspondence to:* Tine Nilsen (tine.nilsen@uit.no)

**Abstract.** The Bayesian hierarchical model BARCAST ("Bayesian Algorithm for Reconstructing Climate Anomalies in Space and Time") climate field reconstruction (CFR) technique, and idealized input data are used in the pseudoproxy experiments of this study. Ensembles of targets are generated from fields of long-range memory stochastic processes using a novel approach. The range of experiment setups include input data with different levels of persistence and levels of proxy noise, but without any form of external forcing. The input data are thereby a simplistic alternative to standard target data extracted from general circulation model (GCM) simulations. Ensemble-based temperature reconstructions are generated, representing the European landmass for a millennial time period. Hypothesis testing in the spectral domain is then used to investigate if the field and spatial mean reconstructions are consistent with either the fractional Gaussian noise (fGn) null hypothesis used for generating the target data, or the autoregressive model of order one (AR(1)) null hypothesis which is the assumed temperature model for this reconstruction technique. The study reveals that the resulting field and spatial mean reconstructions are consistent with the fGn hypothesis for most of the parameter configurations. There are local differences in reconstructed scaling characteristics between individual grid cells, and a generally better agreement with the fGn model for the spatial mean reconstruction than at individual locations. The discrepancy from an fGn is most evident for the high-frequency part of the reconstructed signal, while the long-range memory is better preserved at frequencies corresponding to decadal time scales and longer. Selected experiment setups were found to give reconstructions consistent with the AR(1) model. Reconstruction skill is measured on an ensemble member basis using selected validation metrics. Despite the mismatch between the BARCAST temporal covariance model and the model of the target, the ensemble mean was in general found to be consistent with the target data, while the estimated confidence intervals are more affected by this discrepancy. Our results show that the use of target data with a different spatiotemporal covariance structure than the BARCAST model assumption can lead to a potentially biased CFR reconstruction and associated confidence intervals, because of the wrong model assumptions.

## 1 Introduction

Proxy-based climate reconstructions are major tools in understanding past and predicting future variability of the climate system over a range of timescales. Over the last few decades a considerable progress has been made, and a number of proxy/multiproxy reconstructions of different climate variables have been created. Target regions, spatial density and tem-



poral coverage of the proxy network varied between the studies, with a general trend towards more comprehensive networks and sophisticated reconstruction techniques used. For example, Jones et al. (1998); Moberg et al. (2005); Mann et al. (1998, 2008); PAGES 2k Consortium (2013); Luterbacher et al. (2016); Werner et al. (2017) present reconstructions of surface air temperatures (SAT) for different spatial and temporal domains. The available reconstructions often tend to disagree on aspects

such as timing, duration and amplitude of warm/cold periods, due to different methods, types and number of proxies, and regional delimitation used in the different studies, (Wang et al., 2015). There are also alternative viewpoints on a more fundamental basis considering the level of high frequency versus low frequency variability, see e.g. Christiansen (2011); Tingley and Li (2012). Discrepancies in the Fourier domain can occur among other things due to shortcomings of the reconstruction techniques, such as regression dilution. This describes variance losses back in time and bias of the target variable mean. These

artifacts appear as a consequence of noisy measurements used as predictors in regression techniques based on ordinary least squares (Christiansen, 2011; Wang et al., 2014). The level of high/low frequency variability in reconstructions also depends on the type and quality of the proxy data used as input (Christiansen and Ljungqvist, 2017).

The concept of pseudoproxy experiments was introduced after millennium-long paleoclimate simulations from GCMs first became available, and has been developed and applied over the last decade, (Mann et al., 2005, 2007; Lee et al., 2008). Pseu-

doproxy experiments are used to test the skill of reconstruction methods and the sensitivity to the proxy network used, see Smerdon (2012) for a review. The idea behind idealized pseudoproxy experiments is to extract target data of an environmental variable of interest from long paleoclimate model simulations for an arbitrary reconstruction region. The target data is then sampled in a spatiotemporal pattern that simulates real proxy networks and instrumental data. The target data representing the proxy period is further perturbed with noise to simulate real proxy data in a systematic manner, while the pseudo instrumental

data are left unchanged or only weakly perturbed with noise of magnitude typical for the real world instrumental data. The surrogate pseudoproxy and pseudoinstrumental data are used as input to one or more reconstruction techniques, and the resulting reconstruction is then compared with the true target from the simulation. The reconstruction skill is quantified through statistical metrics, both for a calibration- and a much longer validation interval.

The available pseudoproxy studies have to a great extent used target data from the same GCM model simulations, subsets of

the same spatially distributed proxy network and a temporally invariant pseudoproxy network (Smerdon, 2012). The concept of extracting target data from simpler model simulations has not been widely explored. In the present paper we extend the domain of pseudoproxy experiments. Instead of employing surrogate data from paleoclimate GCM simulations, ensembles of target fields are drawn from a field of stochastic processes with prescribed dependencies in space and time. In the framework of such an experiment design, the idealized temperature field can be thought of as an (unforced) control simulation of the Earth's

surface temperature field with a simplified spatial covariance structure. The primary goal of using these target fields is to test the ability of the reconstruction method to preserve the spatiotemporal covariance structure of the surrogates in the climate field reconstruction. An example of such a study is Werner and Tingley (2015), where idealized target data were generated based on the BARCAST model equations introduced here in Sect. 2.1.

In addition, we test the reconstruction skill on an ensemble member basis using standard metrics including the correlation

coefficient and the root-mean-squared error (RMSE). We also employ the continuous ranked probability score (CRPS), which



is a suitable skill metric for ensemble-based reconstructions, in contrast to the often-used coefficient of efficiency (CE) and reduction of error (RE).

Temporal dependence in a stochastic process over time $t$ is described as persistence or memory, given that the process has a Gaussian probability distribution. A long-range memory (LRM) stochastic process exhibits an autocorrelation function (ACF)

and a power spectral density (PSD) of a power-law form: $C(t) \sim t^{\beta-1}$, and $S(f) \sim f^{-\beta}$ respectively. The power-law behavior of the ACF and the PSD indicates the absence of a characteristic time scale in the time series; the record is *scale invariant* (or just *scaling*). The spectral exponent $\beta$ determines the strength of the persistence. The special case $\beta = 0$ is the white noise process, which has a uniform PSD over the range of frequencies. For comparison, another model often used to describe the background variability of the Earth's SAT is the autoregressive process of order 1 (AR(1)) (Hasselmann, 1976). This process

has a Lorentzian power spectrum and thereby does not exhibit long-range correlations.

For the instrumental time period, studies have shown that detrended local and spatially averaged surface temperature data exhibit long-range memory properties on time scales from months up to decades, (Koscielny-Bunde et al., 1996; Rybski et al., 2006; Fredriksen and Rypdal, 2016). For proxy/multiproxy SAT reconstructions, studies indicate persistence up to a few centuries or millennia, (Rybski et al., 2006; Lovejoy and Schertzer, 2012; Nilsen et al., 2016). The exact strength of persistence

varies between data sets and depends on the degree of spatial averaging, but in general $0 < \beta < 1.3$ is adequate. The value of $\beta > 1$ is usually associated with sea surface temperature, which features stronger persistence due to effects of oceanic heat capacity. The deviation from Gaussianity of instrumental temperatures varies with latitude (Franzke et al., 2012), and the nonlinearity in some types of proxy records also result in nongaussianity (Emile-Geay and Tingley, 2016).

Our basic assumption is that the background temporal evolution of Earth's surface air temperature can be modelled by the

persistent Gaussian stochastic model known as the fractional Gaussian noise (fGn) (Beran et al.)[Chapter 1 and 2], (Rypdal. et al., 2013). This process is stationary, and the persistence is defined by the spectral exponent $0<\beta<1$. The synthetic target data are designed as ensembles of LRM-processes in time, with an exponentially decaying spatial covariance structure. In contrast to using target data from GCM simulations, this gives us the opportunity to vary the strength of persistence in the target data, retaining a simplistic and temporally persistent model for the signal covariance structure. The persistence is varied

systematically to mimic the range observed in actual reconstructions over land, typically $0 < \beta < 1$. The pseudoproxy data quality is also varied by adding levels of white noise corresponding to signal-to-noise ratios by standard deviation (SNR)= $\infty, 3, 1, 0.3$. For comparison, the signal to noise ratio of observed proxy data is normally between 0.5-0.25 (Smerdon, 2012). However, in Werner et al. (2017), most tree-ring series were found to have SNR > 1. Since the target data are represented as an ensemble of independent members generated from the same stochastic process, there is little value in estimating and

analyzing ensemble means from the target and reconstructed time series themselves. Anomalies across the ensemble members will average out, and the ensemble mean will simply be a time series with non-representative variability across scales. Instead we will focus on averages in the spectral sense. The means of the ensemble member-based metrics are used to quantify the reconstruction skill.

The reconstruction method to be tested is the "Bayesian Algorithm for Reconstructing Climate Anomalies in Space and

Time" (BARCAST), based on a Bayesian Hierarchical Model (Tingley and Huybers, 2010a). This is a state-of-the-art paleocli-

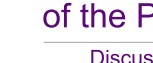

mate reconstruction technique, described in further detail in Sect. 2.1. The motivation for using this particular reconstruction technique in the present pseudoproxy study is the contrasting background assumptions for the temporal covariance structure. BARCAST assumes that the temperature evolution follows an AR(1) process in time, while the target data are generated according to the fGn model. The consequences of using an incorrect null hypothesis for the temporal data structure are illustrated

in Fig. 1. Here, the original time series in Fig. 1a follows an fGn structure. The corresponding power spectrum is plotted in blue in Fig. 1c. Using the incorrect null hypothesis that the data are generated from an AR(1) model, we estimate the AR(1) parameters from the time series in Fig. 1a using Maximum Likelihood estimation. A realization of an AR(1) process with these parameters is plotted in Fig. 1b, with the power spectrum shown in red in Fig. 1c. The characteristic timescale indicating the memory limit of the system is evident as a break in the red AR(1) spectrum. This is an artifact that does not stem from the

original data, but simply occurs because an incorrect assumption was used for the temporal covariance structure.

A particular advantage of BARCAST as a probabilistic reconstruction technique lies in its capability to provide an objective error estimate as the result of generating a distribution of solutions for each set of initial conditions. The reconstruction skill of the method has been tested and compared against a few other CFR techniques using pseudoproxy experiments. Tingley and Huybers (2010b) use instrumental temperature data for North America, and construct pseudoproxy data from some of the

longest time series. BARCAST is then compared to the RegEM method used by Mann et al. (2008, 2009). The findings are that BARCAST is more skillful than RegEM if the assumptions for the method are not strongly violated. The uncertainty bands are also narrower. The other pseudoproxy study is described in Werner et al. (2013), where BARCAST is compared against a CFR method based on canonical correlation analysis (CCA). The pseudo proxies in that paper were constructed from a millennium-long forced run of the NCAR CCSM4 model. The results showed that BARCAST outperformed the CCA method

over the entire reconstruction domain, being similar in areas with good data coverage. There is an additional pseudoproxy study by Gómez-Navarro et al. (2015), targeting precipitation which has a more complex spatial covariance structure than SAT anomalies. In that study, BARCAST was not found to outperform the other methods.

In the following, we describe the methodology of BARCAST and the target data generation in Sect. 2. The spectral estimator used for persistence analyses is also introduced here. Sect. 3 comprises an overview of the experiment setup and explains the

hypothesis testing procedure. Results are presented in Sect. 4 after performing hypothesis testing in the spectral domain of persistence properties in the local and spatial mean reconstructions. The skill metric results are also summarized. Finally, Sect. 5 discuss the implications of our results and provides concluding remarks.

## 2   Data and methods

### 2.1   BARCAST methodology

BARCAST is a climate field reconstruction method, described in detail in Tingley and Huybers (2010a). It is based on a Bayesian hierarchical model with three levels. The true temperature field in BARCAST, $\mathbf{T}_t$ is modelled as a multivariate



first-order autoregressive model (AR(1)) in time. Model equations are defined at the process level:

$$\mathbf{T}_t - \mu \mathbf{1} = \alpha(\mathbf{T}_{t-1} - \mu \mathbf{1}) + \boldsymbol{\epsilon}_t \tag{1}$$

Where the scalar parameter $\mu$ is the mean of the process, $\alpha$ is the AR(1) coefficient, and $\mathbf{1}$ is a vector of ones. The subscript $t$ indexes time in years, and the innovations (increments) $\boldsymbol{\epsilon}_t$ are assumed to be IID normal draws $\boldsymbol{\epsilon}_t \sim N(0, \boldsymbol{\Sigma})$, where

$$5 \quad \Sigma_{ij} = \sigma^2 \exp(-\phi|\mathbf{x}_i - \mathbf{x}_j|) \tag{2}$$

is the spatial covariance matrix depicting the covariance between locations $\mathbf{x}_i$ and $\mathbf{x}_j$.

The spatial e-folding distance is $1/\phi$ and is chosen to be $\sim 1000$ km for the target data. This is a conservative estimate resulting in weak spatial correlations for the variability across a continental landmass. (North et al., 2011) estimate that the decorrelation length for a 1-year average of Siberian temperature station data is 3000 km. On the other hand, Tingley and Huybers (2010a) estimate a decorrelation length of 1800 km for annually mean global land data. They further use annual mean instrumental and proxy data from the North American continent to reconstruct SAT back to 1850, and find a spatial correlation length scale of approximately 3300 km for this BARCAST reconstruction. Werner et al. (2013) use $1/\phi \sim 1000$ km as the mean for the lognormal prior in the BARCAST pseudoproxy reconstruction for Europe, but the reconstruction has correlation lengths between 6000-7000 km. The reconstruction of Werner et al. (2017) has spatial correlation length slightly longer than 1000 km.

On the data level, the observation equations for the instrumental and proxy data are:

$$20 \quad \mathbf{W}_t = \begin{pmatrix} \mathbf{H}_{I,t} \\ \beta_1 \cdot \mathbf{H}_{P,t} \end{pmatrix} \mathbf{T}_t + \begin{pmatrix} \mathbf{e}_{I,t} \\ \mathbf{e}_{P,t} + \beta_0 \mathbf{1} \end{pmatrix} \tag{3}$$

Where $\mathbf{e}_{I,t}$ and $\mathbf{e}_{P,t}$ are multivariate normal draws $\sim N(0, \tau_I^2 \mathbf{I})$ and $\sim N(0, \tau_P^2 \mathbf{I})$. $\mathbf{H}_{I,t}$ and $\mathbf{H}_{P,t}$ are selection matrices of ones and zeros which at each year select the locations where there are instrumental/proxy data. $\beta_0$ and $\beta_1$ are parameters representing the scaling factor and bias of the proxy records relative to the temperatures. Note that these two parameters have no relation to the spectral parameter $\beta$. The BARCAST parameters are distinguished by their indices, the notation is kept as it is to comply with existing literature.

The remaining level is the prior. Weakly informative but proper prior distributions are specified for the scalar parameters and the temperature field for the first year in the analysis. The priors for all parameters except $\phi$ are conditionally conjugate. The Markov-Chain Monte Carlo (MCMC) algorithm known as the Gibbs sampler (with one Metropolis step) is used for the posterior simulation. Table A1 sums up the prior distributions and the choice of hyperparameters for the scalar parameters in



BARCAST. The CFR version applied here has been updated as described in Werner and Tingley (2015). The updated version allows inclusion of proxy records with age uncertainties. This property will not be used here directly, but it implies that proxies of different types may be included. Instead of estimating one single parameter value of $\tau_P^2$, $\beta_0$ and $\beta_1$, the updated version estimates individual values of the parameters for each proxy record (Werner et al., 2017).

In the present study, the Metropolis-coupled MCMC algorithm is run for 5000 iterations, running three chains in parallel. Each chain is assumed equally representative for the temperature reconstruction if the parameters converge. There are a number of ways to investigate convergence, for instance one can study the variability in the plots of draws of the model parameters as a function of step number of the sampler, as in Werner et al. (2013). However, a more robust convergence measure can be achieved when generating more than one chain in parallel. By comparing the within-chain variance to the between-chain
variance we get the convergence measure $\hat{R}$, (Gelman et al., 2003, Chapter 11). $\hat{R}$ close to one indicates convergence for the scalar parameters.

There are numerous reasons why the parameters may fail to converge, including inadequate choice of prior distribution and/or hyperparameters or using an insufficient number of iterations in the MCMC algorithm. It may also be problematic if
the spatiotemporal covariance structure of the observations or surrogate data deviate strongly from the model assumption of BARCAST.

Since BARCAST is a probabilistic reconstruction technique it was used to generate an ensemble of reconstructions, in order to achieve a mean reconstruction as well as uncertainties. In our case, the draws for each temperature field and parameter are thinned so that only every 10 of the 5000 iterations are saved; this secures independence of the draws.

The output temperature field is reconstructed also in grid cells without observations, which is a unique property compared to other well-known field reconstruction methods such as the regularized expectation maximum technique (RegEM) applied in Mann et al. (2009). Note that the assumptions for BARCAST should generally be different for land and oceanic regions, due to the differences in characteristic timescales and spatiotemporal processes. BARCAST is so far only configured to deal with continental land data, (Tingley and Huybers, 2010a).

## 2.2 Target data generation

While generating ensembles of synthetic LRM processes in time is straightforward using statistical software packages, it is more complicated to generate a field of persistent processes with prescribed spatial covariance. Below we describe a novel technique that fulfills this goal, which can be extended to include more complicated spatial covariance structures. Such a spatiotemporal field of stochastic processes may potentially have many theoretical and practical applications.

Generation of target data begins with reformulating eq. (1) so that the temperature evolution is defined from a power-law function instead of an AR(1). The continuous-time version of Eq. (1) (with $\mu = 0$) is the ordinary differential equation:

$$\frac{d\mathbf{T}}{dt} = -(1-\alpha)\mathbf{I}\mathbf{T} + \boldsymbol{\epsilon}_t, \qquad \text{Where } \mathbf{I} \text{ Is the identity matrix.} \qquad (4)$$



with the solution:

$$\mathbf{T}(t) = \int_0^t \exp^{-(1-\alpha)\mathbf{I}(t-s)} \boldsymbol{\epsilon}_s ds \tag{5}$$

The exponential kernel is then replaced by a power-law function to yield:

$$5 \quad \mathbf{T}(t) = \int_0^t (t-s)^{\beta/2-1} \boldsymbol{\epsilon}_s ds \tag{6}$$

This expression describes the long-memory response to the noise forcing after time $t = 0$. Note that there is no contribution from the initial condition $\mathbf{T}(0)$. This is because $\mathbf{T}(t)$ in Eq. (6) in contrast to Eq. (5) is no longer a solution to an ordinary differential equation, but rather a fractional differential equation, whose solution for $t > 0$ depends not only on the initial condition but the entire time history of $\mathbf{T}(t)$, $t \in (-\infty, 0)$. Eq. 6 effectively corresponds to neglecting the contribution from the noisy forcing prior to $t = 0$.

In discrete form, the convolution integral in Eq. 6 is approximated over an index $s$:

$$\sum_{s=0}^{t} (t-s+\tau_0)^{\beta/2-1} \boldsymbol{\epsilon}_{\mathbf{s}} \tag{7}$$

Note that here the stabilizing term $\tau_0$ is added to avoid the singularity at $s = t$. The optimal choice would be to choose $\tau_0$ such that the term in the sum arising from $s = t$ represents the integral in the interval $s \in (t-1, t)$, i.e.,

$$\tau_0 = \int_0^{\tau_0} \tau^{\beta/2-1} d\tau,$$

which has the solution $\tau_0 = \beta/2$.

Summation for time steps $s, t = 1, 2, ....N$ and $\tau = t - s$ of (7) results in the matrix $\mathbf{G}$ with terms:

$$15 \quad \mathbf{G}(\tau) = (\tau + \beta/2)^{(\beta/2-1)} \Theta(\tau) \tag{8}$$

Where $\Theta(\tau)$ is the unit step function. $\boldsymbol{\epsilon}_t$ is kept identical as in eq. (1) and (2). The target temperature field $\mathbf{T}$ at time $t$ can be calculated as:

$$\mathbf{T}_t = \mathbf{G}_t \boldsymbol{\epsilon}_t \tag{9}$$





## 2.3 Estimation of power-spectral density

The temporal dependencies in the reconstructions are investigated to obtain detailed information about how the reconstruction technique may alter the level of variability on different scales, and how sensitive it is to the proxy data quality. Persistence properties of target data, pseudoproxies and the reconstruction are compared and analyzed in the spectral domain using the

periodogram as the estimator.

The periodogram is defined here in terms of the discrete Fourier transform $H_m$ as $S(f_m) = (2/N)|H_m|^2, m = 1, 2, \ldots, N/2$. The sampling time is an arbitrary time unit, and the frequency is measured in cycles per time unit: $f_m = m/N$. $\Delta f = 1/N$ is the frequency resolution and the smallest frequency which can be represented in the spectrum.

Power spectra are visualized in log-log plots since the spectral exponent can be estimated by a simple linear fit to the

spectrum. The raw and log-binned periodograms are plotted, and $\beta$ is estimated from the latter. Log-binning of the periodogram is used here for analytical purposes, since it is useful with a representation where all frequencies are weighted equally with respect to their contributions to the total variance.

It is also possible to use other estimators for scaling analysis, such as the detrended fluctuation analysis (DFA, Peng et al. (1994)), or wavelet variance analysis (Malamud and Turcotte, 1999). Each estimation technique has benefits and deficiencies,

and one can argue for the superiority of methods other than PSD or the use of a multi-method approach. However, we consider the spectral analysis to be adequate for our purpose and refer to Nilsen et al. (2016) for a discussion on selected estimators for scaling analysis.

## 3   Experiment setup

The experiment domain configuration is selected to resemble that of the continental landmass of Europe, with $N = 56$ grid cells

of size 5°x5°. The reconstruction period is 1000 years, reflecting the last millennium. The reconstruction region and period are inspired by the BARCAST reconstructions in Werner et al. (2013); Luterbacher et al. (2016) and approximate the density of instrumental and proxy data in reconstructions of the European climate of the last millennium. The temporal resolution for all types of data is annual. By construction the target fGn data are meant to be an analogue of the unforced SAT field and hence can be considered as representing GCM control simulations. We will study both the field and spatial mean reconstruction.

Pseudoinstrumental data cover the entire reconstruction region for the time period 850-1000 and are identical to the noise-free values of the true target variables. The spatial distribution of the pseudoproxy network is highly idealized as illustrated in Fig. 2, the data covers every fourth grid cell for the time period 1-1000. The pseudoproxies are constructed by perturbing the target data with white noise according to Eq. 3. The variance of the proxy observations is $\tau_P^2$, and the SNR is calculated as:

$$\text{SNR} = \frac{\beta_1^2 \text{Var}(T_t)}{\tau_P^2} \tag{10}$$

Our set of experiments is summarized in Table 1 and comprises target data with three different strengths of persistence, $\beta = 0.55, 0.75, 0.95$ and pseudoproxies with four different signal to noise ratios by standard deviation (SNR): SNR=$\infty, 3, 1,$





and 0.3. In total, 20 realizations of target pseudoproxy and pseudoinstrumental data are generated for each combination of $\beta$ and SNR and used as input to BARCAST. The reconstruction method is probabilistic and generates ensembles of reconstructions for each input data realization. In total, 30 000 ensemble members are constructed for every parameter setup.

## 3.1 Hypothesis testing

Hypothesis testing in the spectral domain is used to determine which pseudoproxy/reconstructed data sets can be classified as fGn with the prescribed scaling parameter, or as AR(1) with parameter $\alpha$ estimated from BARCAST. The power spectrum for each ensemble member of the local/spatial mean reconstructions is estimated, and the mean power spectrum is then used for further analyses. The first null hypothesis is that the data sets under study can be described using an fGn with the prescribed scaling parameter for the target data at all frequencies, $\beta_{\text{target}} = 0.55, 0.75$ and $0.95$ respectively. For testing we generate a

Monte Carlo (MC) ensemble of fGn series with a value of the scaling parameter identical to the target data. The power spectrum of each ensemble member is estimated, and the confidence range for the theoretical spectrum is then calculated using the 2.5 and 97.5 quantiles of the log-binned periodograms of the MC ensemble. The null hypothesis is rejected if the log-binned mean spectrum of the data is outside of the confidence range for the fGn model at any point.

   The second null hypothesis tested is that the data can be described as an AR(1) process at all frequencies, with the parameter

$\alpha$ estimated from BARCAST. Distributions for all scalar parameters including the AR(1) parameter $\alpha$ are provided through the reconstruction algorithm. The mean of this parameter was used to generate a Monte Carlo ensemble of AR(1) processes. The MC ensemble and the confidence range is then based on log-binned periodograms for this theoretical AR(1) process.

   Figure 3 presents an example of the hypothesis testing procedure. The fGn 95% confidence range is plotted as a shaded gray area in the log-log plot together with the mean raw and mean log-binned periodograms for the data to be tested. Blue curve

and dots represent mean raw and log-binned PSD for pseudoproxy data, red curve and dots represent mean raw and log-binned PSD for reconstructed data. The gray, dotted line is the ensemble mean.

   Note that the formulation of the two null hypotheses gives no restriction about the normalization of the fGn and AR(1) data used to generate the MC ensemble. Particularly, they do not have to be standardized in the same manner as the pseudoproxy/reconstructed data. This makes the experiment more flexible, as the spectral confidence range of the MC ensemble can

be shifted vertically to better accommodate the data under study. A standard normalization of data includes subtracting the mean and normalizing by the standard deviation. This was sufficient to support the null hypotheses in many of our experiments. A different normalization could also be used, for instance if one considers only the high- or only the low frequency variability to be representative of the true variability in the time series. This is often the case with proxy-based reconstructions. In this case, it would be useful to calculate the variance in the high- or low frequency range by filtering the data, and then normalize

the unfiltered data by this variance.





## 4 Results

BARCAST successfully estimates posterior distributions for all reconstructed temperature fields and scalar parameters. Convergence is reached for the scalar parameters despite the inconsistency of the input data temporal covariance structure with the default assumption of BARCAST. Table A2 lists the true parameter values used for the target data generation, and Tab.

A3 summarizes the mean of the posterior distributions estimated from BARCAST. Studying the parameter dependencies, it is clear that the posterior distributions of $\alpha$ and $\sigma^2$ depend on the prescribed $\beta$ and to a lesser extent, SNR for the target data. The mean values of the $\alpha$ distributions were used to generate Monte Carlo ensembles of AR(1) processes for hypothesis testing. For the parameters $\tau_P^2$, $\beta_0$ and $\beta_1$, there are individual posterior distributions for each of the local proxy records. Instead of listing the posterior distributions of $\tau_P^2$ and $\beta_1$ we have estimated the local reconstructed SNR at each proxy location using Eq.

10  10.

   Further results concern the spectral analyses and skill metrics. For each ensemble member of the input dataset and temperature reconstruction, the PSD is estimated and the mean spectrum is used in further analyses. All references to spectra in the following correspond to mean spectra. Analyses of the reconstruction skills presented below are performed on a grid point basis as well as for the spatial mean reconstruction. While the latter provides an aggregate summary of the method's

ability to reproduce specified properties of the climate process on a global scale, the former evaluates the BARCAST spatial performance.

### 4.1 Isolated effects of added proxy noise on scaling properties in the input data

The scaling properties of the input data are modified already when the target data are perturbed with white noise to generate pseudoproxies. The power spectra shown in blue in Fig. 3 are used to illustrate these effects for one arbitrary proxy location and

$\beta$. Figure 3a shows the spectrum for SNR= $\infty$, which is the unperturbed fGn signal corresponding to ideal proxies. Panels 3b, c and d show spectra for SNR=3, 1 and 0.3 respectively. The effect of added white noise in the spectral domain is manifested as flattening of the-high frequency part of the spectrum equal to $\beta = 0$, and a gradual transition to higher $\beta$ for lower frequencies. The pseudoproxies in panels 3b, c and d all deviate from the confidence range on the highest frequencies, while the log-binned spectrum in Fig. 3(d) is outside on lower frequencies as well. The hypothesis testing results for $\beta_{\text{target}} = 0.55$ and $0.95$ are the

same.

### 4.2 Memory properties in the field reconstruction

Hypothesis testing was performed in the spectral domain for the field reconstructions, with the two null hypotheses formulated as follows:

1: The reconstruction is consistent with the fGn structure in the target data for all frequencies.
   2: The reconstruction is consistent with the AR(1) model used in BARCAST for all frequencies.



Table 2 summarizes the results for all experiment configurations at local grid cells, both directly at and between proxy grid cells. Figures are provided only for one $\beta$, all SNR. Figure 3 shows the mean power spectra generated for the experiment

$\beta = 0.75$ at one arbitrary proxy grid cell of the reconstruction in red. The fGn model is adequate for SNR=∞, 3 and 1, shown in panel 3a-c. For the lowest SNR presented in panel d, the reconstruction spectrum falls outside the confidence range of the theoretical spectrum for one single log-binned point. Not unexpectedly, the difference in shape of the PSD between the pseudoproxy and reconstructed spectra increases with decreasing SNR. The difference is largest for the noisiest proxies with SNR=0.3. This figure does not show the hypothesis testing for the reconstructed spectrum using the AR(1) null hypothesis.

Results show that this null hypothesis is rejected for all cases except SNR=0.3.

The hypothesis testing results vary moderately between the individual grid cells. PSD analyses of the local reconstructions using the same $\beta$ but in an arbitrary non-proxy location are displayed in Fig. 4. Here, the reconstructed mean spectrum is plotted in gray together with both the fGn 95% confidence range (blue) and the AR(1) confidence range (red). Hypothesis testing using null hypothesis 1 and 2 is performed systematically. Wherever the reconstructed spectrum is consistent with the

fGn/AR(1) model, the edges of the associated confidence range are plotted with solid lines. We find that all reconstructed spectra are consistent with the AR(1) model, while only the cases SNR=∞ and 3 are consistent with the fGn model.

### 4.3    Memory properties in the spatial mean reconstruction

The spatial mean reconstruction is calculated as the mean of the local reconstructions for all grid cells considered, weighted by the areas of the grid cells. The reconstruction region considered is $37.5° - 67.5°$N, $12.5° - 47.5°$E. Figure 5 shows the raw

and log-binned periodogram of the spatial mean reconstruction for $\beta_{target} = 0.75$ in gray, together with the 95% confidence range of fGn generated with $\beta = 0.75$ (blue) and AR(1) confidence range (pink). All hypothesis testing results for the spatial mean reconstruction are summarized in Table 3. Results show that the fGn null hypothesis is suitable for all values of $\beta$ and SNR, while the AR(1) null hypothesis is also supported for the case $\beta = 0.55, 0.75$, SNR=0.3.

### 4.4    Assessment of reconstruction skill

It is common practice in paleoclimatology to evaluate reconstruction skill using metrics such as the Pearson's correlation coefficient, the root-mean squared error (RMSE), and the coefficient of efficiency (CE) (Smerdon et al., 2011; Wang et al., 2014). However, the CE metric is improper for reconstructions based on the Bayesian framework (Werner et al., 2017), and will not be used. Instead we will use the continuous ranked probability score (CRPS), (Gneiting and Raftery, 2007), which was earlier used in (Werner and Tingley, 2015; Werner et al., 2017). Skill values are estimated on an ensemble member basis, but

results given below are mean values for the entire ensemble.

The skill of the reconstruction method is measured using the RMSE, the Pearson's correlation coefficient ($r$) and the CRPS. Since the CRPS is less well-known than the two former methods, we define it briefly and refer to Gneiting and Raftery (2007)





for further details.

$$\mathrm{CRPS} = \frac{1}{2}\mathbb{E}_F|X - X'| - \mathbb{E}_F|X - x| \tag{11}$$

Where $F$ refer to the cumulative distribution function. X and X' are ensemble members of the reconstruction, and x is the target variable. The first term represents the mismatch between the cumulative distribution functions of all pairs of ensemble member reconstructions, while the second term measures the mismatch between all ensemble members and the target cumulative distribution function. The CRPS score is given for each individual time step. The estimates are given in the same unit as the variable under study, here surface temperature. In the following we are handling the two subcomponents of the CRPS: the temporally averaged score metric, called the average potential CRPS, ($\overline{\mathrm{CRPS}}_{\mathrm{pot}}$) and the Reliability, representing the validity of the uncertainty bands (Hersbach, 2000). The $\overline{\mathrm{CRPS}}_{\mathrm{pot}}$ metric is akin to the Mean Absolute Error of a deterministic forecast. The Reliability metric tests whether for all cases in which a certain probability p was forecast, on average, the event occurred with that fraction p. Or, using other words, it is tested whether the ensemble is capable of generating cumulative distributions that have, on average, the desired statistical property. Note that a perfectly reliable system has Reliability=0.

### 4.4.1 Skill measure results

The figures 6, 7, 8 and 9 display the spatial distribution of the ensemble mean skill metrics for the experiment $\beta$=0.75 and all noise levels. All figures show a spatial pattern of dependence on the proxy availability, with the best skill at proxy sites except in Fig. 9. Figure 6 shows the local correlation coefficient $r$ between the target and the localized reconstruction for the verification period 1-1849. The correlation is highest for the ideal-proxy experiment in Fig.6a, and gradually decreases at all locations as the noise level rises in panels b-d. Fig. 7 shows the local RMSE. Note that Fig. 7 use the same color bar as in Fig. 6, but best skill is achieved where the RMSE is low. Fig. 8 shows the distribution of $\overline{\mathrm{CRPS}}_{\mathrm{pot}}$. The minimum estimate for the $\overline{\mathrm{CRPS}}_{\mathrm{pot}}$ at proxy locations in Fig. 8a is 0.15, which indicates a low error between the temporally averaged reconstruction and the target. For the remaining locations in Fig. 8a-d, the estimates are between 0.61-0.67. The temperature unit has not been given for our pseudoproxy reconstructions, but for real-world reconstructions the unit will typically be degrees Celsius (°C) or Kelvin (K). The Reliability shown in Fig. 9 is generally low if the proxy locations in 9a are neglected. Except for these grid cells, the local Reliability score ranges between $1*10^{-3}$ to 0.32, with lowest (best skill) estimate for the lowest SNR (strongest noise). The improved Reliability for higher noise scenarios is apparently due to a better consistency between the BARCAST model assumption and the LRM signal which is deteriorated with a high additive noise level. The maximum Reliability score at the proxy locations in 9a is 0.93, which indicates poor reconstruction skill when the validity bands are considered. For these locations, the contrasting skill scores obtained for the $\overline{\mathrm{CRPS}}_{\mathrm{pot}}$ and the Reliability indicate that the reconstructions are on average in good agreement with the target, but the confidence range is not.

Table 4 summarizes the mean local skill for all experiments and skill metrics. BARCAST is in general able to reconstruct major features of the target field. A general conclusion that can be drawn is that the skill metrics vary with SNR, but are less sensitive to the value of $\beta$. For the highest noise-level SNR=0.3, the values obtained for $r$ and the RMSE are in line with those listed in Table 1 of Werner et al. (2013).



Table 5 sums up the ensemble mean skill values of $r$ and RMSE for the spatial mean reconstructions. The skill values are considerably higher than for the local field reconstructions. The CRPS scores have not been evaluated for the spatial mean reconstruction.

## 5 Discussion

In this study we have tested the capability of BARCAST to preserve temporal long-range memory properties of reconstructed data. Pseudoproxy and pseudoinstrumental data were generated with a prescribed spatial covariance structure and LRM temporal persistence using a new method. The data were then used as input to the BARCAST reconstruction algorithm, which by construction uses an AR(1) model for temporal dependencies in the input/output data. The spatiotemporal availability of observational data was kept the same for all experiments in order to isolate the effect of the added noise level and the strength of persistence in the target data. The mean spectra of the reconstructions are tested against the null hypotheses that the recon-

structed data can be represented as LRM processes using the parameters specified for the target data, or as AR(1) processes using the parameter estimated from BARCAST. We found that despite the default assumptions in BARCAST, not all local and spatial mean reconstructions were consistent with the AR(1) model. Typically, the local reconstructions at grid cells between proxy locations are shown to follow the AR(1) model, while the local reconstructions directly at proxy locations are more

similar to the original fGn data. However, the parameter setup is crucial for the spectral shape of these local reconstructions, with higher noise levels indicating better agreement with the AR(1) model than the fGn.

Moreover, the hypothesis testing results show that all spatial mean reconstructions are consistent with the fGn null hypothesis. For the two cases $\beta = 0.55, 0.75$, SNR=0.3, the reconstructions are also consistent with the AR(1) null hypothesis.

The fact that the reconstructed LRM properties are better preserved directly at proxy locations than between proxies is an

expected result. By construction, BARCAST estimates the posterior distributions of $\tau_P^2$ and $\beta_1$, which are related to the signal to noise ratio through Eq. 10. Tab. A3 shows that the estimated reconstructed SNR is close to the true SNR. The reconstruction is designed to follow the model equations 1 and 2 of the true temperature field. The temperatures between proxies therefore have to be generated through stochastic infilling using all the estimated parameters, while directly at proxy sites the only necessary operation is to remove the assumed proxy noise.

Due to the interdependence of the BARCAST parameters, the increase in the estimated AR(1) parameter $\alpha$ and $\beta_0$ is accompanied by a decrease of $\sigma^2$ for noisy input data. These erroneous estimates influence the resulting reconstruction.

The power spectra in Fig. 3, 4 and 5 show that the temporal covariance structure of the reconstructions is altered compared with the target data for all experiments where noisy input data were used. Furthermore, the spectra of the pseudoproxies and the reconstructions in 3b-d all deviate from the target in the high frequency range, but for different reasons. The pseudoproxy data

deviate from the target due to the white proxy noise component, while the reconstruction deviate because BARCAST quantifies the proxy noise from an AR(1) assumption. This has important implications for how paleoclimate reconstructions should be interpreted. Real-world proxy data are generally noisy, and the noise level is normally at the high end of the range studied here. We demonstrate that the variability-level of the reconstructions does not exclusively reflect the characteristics of the target





data, but is also influenced by the fitting of data to a model that is not necessarily correct. Other reconstruction techniques that may experience similar deficiencies is the regularized expectation-maximization algorithm (RegEM), (Schneider, 2001; Mann et al., 2007), and all related models (CCA, PCA), that assumes observations at subsequent years are independent (Tingley and Huybers, 2010b).

Our results further suggest that the spatial mean reconstructions are to a small extent more consistent with the LRM null hypothesis than local values. This is clear from the spatial mean reconstruction spectra (gray curves in Fig. 5) and from comparing the hypothesis testing results in Table 2 and 3. The improvement in scaling behavior is expected, as the small-scale variability denoted by $\epsilon_t$ in Eq. 9 is averaged out. Eliminating local disturbances naturally results in a more coherent signal. However, the spatial mean of the target data set does not have a significant higher $\beta$ than local target values. This is due to

the relatively short spatial correlation length chosen: $1/\phi = 1000$ km. In observed temperature data, spatial averaging tends to increase the scaling parameter $\beta$ (Fredriksen and Rypdal, 2016).

If the first null hypothesis used here was modified so that only the low-frequency components of the spectra were required to fall within the confidence ranges, more of the reconstructions would be consistent with the fGn model. After all, we have the information that the high-frequency component is affected by noise, and we know the color and level of this noise. However,

from studying the spectra in Fig. 3, 4 and 5, it is generally unclear where one should set a threshold, since the spectra show a gradual change with a lack of any abrupt breaks. Considering real-world proxy records, the noise color and level is generally unknown. We know there are certain sources of noise that are not related to climate influencing different frequency ranges. However, it is difficult to decide when the noise becomes negligible compared with the effects of climate driven processes. The decision to use all frequencies for the hypothesis testing in this idealized study is therefore a conservative and objective choice.

The skill metrics used to validate the reconstruction skill are the RMSE, $r$ and CRPS, the latter divided into the $\overline{\mathrm{CRPS}}_{\mathrm{pot}}$ and the Reliability. We stress that even though the estimates of RMSE, correlation and $\overline{\mathrm{CRPS}}_{\mathrm{pot}}$ indicate skillful mean reconstructions, this does not necessarily imply a reliable reconstruction in terms of correct confidence intervals. The Reliability reflects this uncertainty, which is an important measure for probabilistic reconstruction techniques.

The power spectra can also be used to gain information about the fraction of variance lost/gained in the reconstruction

compared with the target. This fraction is in some sense the bias of the variance, and was found by integrating the spectra of the input and output data over frequency. The spatial mean target/reconstructions were used, and the mean log-binned spectra. The total power in the spatial mean reconstruction and the target were estimated, and the ratio of the two provides the under/overestimation of the variance: $\mathrm{R}_{\mathrm{Var}} = \frac{\mathrm{Var}(T_{t\,(\mathrm{rec})})}{\mathrm{Var}(T_{t\,(\mathrm{target})})}$. A ratio less than unity implies that the reconstructed variance is underestimated compared with the target. Our analyses for the total variance reveal that the ratio varies between 0.83-1.05 for

the different experiments and typically decreases for increasing noise levels. How much the ratio decreases with SNR depends on $\beta$, with higher ratios for higher $\beta$ values. For example, R$\sim 1$ for all $\beta$, SNR$= \infty$ and progressively decreases to R=0.83, 0.89 and 0.94 for SNR=0.3, $\beta = 0.55, 0.75, 0.95$ respectively. In other terms, there are larger variance losses in the reconstruction for smaller values of $\beta$ than for higher $\beta$. We also divided the spectra into three different frequency ranges as shown in Fig. 10 to test if the fraction of variance lost/gained is frequency-dependent. The sections separate low frequencies corresponding

approximately to centennial timescales, mid frequencies corresponding to timescales between decades and centuries, and high





frequencies corresponding to timescales shorter than decadal. The results show no systematic differences between the frequency ranges associated with the parameter configuration.

Previously, the scaling properties of millennium-long paleoclimate reconstructions have been studied in e.g. Lovejoy and Schertzer (2012); Nilsen et al. (2016). These papers present different viewpoints on scaling models used to represent Earth's

surface temperature variability on a range of timescales. Lovejoy and Schertzer (2012) suggests that climate variability on timescales from months to centuries can be denoted "Macroweather" and described using a scaling parameter $\beta \sim 0.2$, while variability on centennial timescales and longer is "Climate" with $\beta \sim 1.4$. This concept involving a separation of scaling regimes around centennial timescales was challenged by Nilsen et al. (2016). It was demonstrated that a spread in scaling parameters follows naturally from analyzing a range of proxy-based reconstructions for the Holocene covering different spatial

regions and dynamical regimes. The occurrence of a second scaling regime was exclusively observed when analyzing time-series including the last glacial period, which are nonstationary and involves nonlinearities that are not present for the Holocene climate. In the present paper it has been shown that both proxy noise and the BARCAST reconstruction technique contribute to alteration of the memory properties of the reconstructed data, introducing artifacts that may be interpreted as scale-breaks. None of these effects are intrinsic to the target data signal, but are introduced through non-climatic effects. Observing only the

reconstruction may not give the complete answer on the temporal structure of the true temperature signal.

## 5.1   Implications for real proxy data

The spectral shape of the input pseudo proxy data plotted in blue in Fig. 3 are similar to spectra of observed proxy data as observed in e.g. some types of tree-ring records, (Franke et al., 2013; Zhang et al., 2015; Werner et al., 2017). Franke

et al. (2013); Zhang et al. (2015) found that the scaling parameters $\beta$ were higher for tree-ring based reconstructions than for the corresponding instrumental data for the same region. Werner et al. (2017) present a new spatial SAT reconstruction for the Arctic, using the BARCAST methodology. The reconstruction is based on annually layered records and layer counted archives with age uncertainties. The persistence properties of the input proxy records and the reconstructed temperatures were investigated using the same spectral techniques as here. Fig. A4 in Werner et al. (2017) presents a map over the Arctic and an

overview of the spatial distribution and type of proxy record. It also indicates if the proxy record is consistent with an AR(1) process null hypothesis or an fGn based on hypothesis testing. The analyses demonstrated that several of the tree-ring records could not be categorized as neither AR(1) or scaling processes, but featured spectra similar to the pseudoproxy spectrum in Fig. 3c-d. The characteristic flat spectrum at high frequencies, and the increased power on bidecadal frequencies and lower can give the impression that the low-frequency power is inflated. However, from the presented experiments we know it is rather

the high frequency power that is affected by the added white noise. We hypothesize that the possible mechanism(s) altering the variability can be due to effects of the tree-ring processing techniques, specifically the methods applied to eliminate the biological tree aging effect on the growth of the trees (Briffa et al., 1992). The actual tree ring width is a superposition of the age-dependent curve, which is individual for a tree, and a signal that can often be associated with climatic effects on the tree growth process. To correct for the biological age-effect, the raw tree-ring growth values are often transformed into proxy




indices using the Regional Curve Standardization technique (RCS, Briffa et al. (1992)). This technique attempts to eliminate biological age effects on tree-growth while preserving low frequency variability. As an example, consider tree-ring width as a function of age (Helama et al., 2017). For a number of individual tree-ring records, each record is aligned according to their biological years. The mean of all the series is then modelled as a negative exponential function (the RCS curve). To construct

the RCS chronology, the raw, individual tree-ring width curves are divided by the mean RCS curve for the full region. The RCS chronology is then the average of the index individual records. It is likely that the shape of a particular tree-ring width spectrum reflects the uncertainty in the RCS curve, which is expected to be largest at the timescales corresponding to the initial stage of a tree growth, where the slope of the growth curve is generally steeper (i.e. of the order of a few decades). In particular, there may be slightly different climate processes affecting the growth of different trees, causing localized nonlinearities that

limit the representativeness of the derived exponential RCS. We therefore suggest that the observed excess of LRM properties in some of the tree ring-based proxy records could be an artifact of the fitting procedure.

### 5.2   Concluding remarks

A natural continuation of the pseudoproxy study presented here would be to generate target data using a more complex model. The stochastic-diffusive models described in North et al. (2011); Rypdal et al. (2015) make interesting candidates because of the

alternative method for generating spatial covariance. The reconstruction technique used in this paper generates a signal without spatial dynamics, where the spatial covariance is defined through the noise term. On the other hand, the stochastic-diffusive models generate the spatial covariance through the diffusion, without spatial structure in the noise term. The latter model type may be considered more physically correct and intuitive than the simplistic model used here. North et al. (2011) use an exponential model for the temporal covariance structure, while Rypdal et al. (2015) use an LRM model. However, the intention

of the present study was to conduct experiments where the target data follows all the model assumptions of BARCAST, except for the temporal correlation structure. Since this small modification had a pronounced effect on the reconstructions it is likely that using a different model would have even larger influence. Using the stochastic diffusive models in either North et al. (2011) or Rypdal et al. (2015) it would also be possible to implement external forcing and responses to these forcings in the target data to make the numerical experiments more realistic.

Another extension proposed for BARCAST already in Tingley and Huybers (2010a) was to generalize the spatial covariance structure using the Matérn covariance function. This would make the assumptions of BARCAST more realistic with respect to teleconnections. The change has been implemented but slows the algorithm down substantially in its present form.

Smerdon (2012) further proposed a number of improvements to be implemented in future pseudoproxy experiments, including accounting for temporal nonuniform availability of proxy data. Most studies overestimate the proxy sampling in the earlier

part of the reconstruction period when using temporally invariant pseudoproxy networks. BARCAST is fully capable of taking such networks as input. Wang et al. (2014) later performed pseudo proxy experiments using four different CFR techniques, and tested two types of proxy networks: one that is fixed through the entire reconstruction period and one where the number of proxy records is reduced back in time, (staircase network). The effect of temporal heterogeneities is unexpectedly that the



reconstruction skill does not decrease strictly following the proxy availability. Strong forcing events has a larger impact on the skill according to that study.

The pseudoproxy study presented here is based on simplistic target data generated using a novel technique. The generation of the input data requires far less computation power and time than for GCM paleoclimatic simulations, but also results in less realistic target temperature fields. However, we demonstrate that there are many areas of use for these types of data, including statistical modelling and hypothesis testing. In particular, the pseudoproxy experiment presented here may be replicated using different index or field reconstruction techniques.

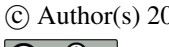



## Appendix A: Information on true parameters, prior and posterior distributions of BARCAST parameters

The forms of the prior PDF's for the scalar parameters in BARCAST are identical to those used in (Werner et al., 2013). The values of the hyperparameters were chosen after analyzing the target data. The forms of the priors and the values of the hyperparameters are listed in Tab.A1.

The parameter values prescribed for the target data are listed in Tab.A2. The instrumental observations are identical to the true target values, and the instrumental error variance $\tau_I^2$ is therefore zero. The proxy noise variance $\tau_P^2$ is varied systematically for the different SNR through the relation in Eq. 10

10    The mean of the posterior distributions of the BARCAST parameters $\alpha, \mu, \sigma^2, 1/\phi$, $\tau_I^2$ and $\beta_0$ are listed in Tab.A3, together with the reconstructed SNR.

*Competing interests.*  The authors declare that they have no conflict of interest.

*Acknowledgements.*  T.N. was supported by the Norwegian Research Council (KLIMAFORSK program) under grant no. 229754, and partly by Tromsø Research Foundation via the UiT project A31054.

15   D. V. D was partly supported by Tromsø Research Foundation via the UiT project A33020.
J.P.W. gratefully acknowledges support from the Centre for Climate Dynamics (SKD) at the Bjerknes Centre.
D.D.V., T.N. and J.P.W. also acknowledge the IS-DAAD project 255778 HOLCLIM for providing travel support.





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





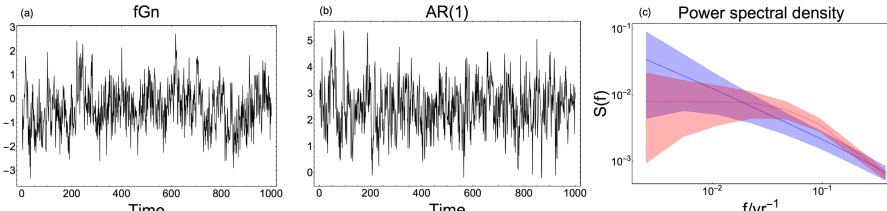

**Figure 1.** (a) Arbitrary fGn time series with $\beta = 0.75$. (b) Arbitrary time series of an AR(1) process with parameters estimated from the time series in (a) using Maximum Likelihood. (c) Log-log spectrum showing 95% confidence ranges based on Monte Carlo ensembles of fGn with $\beta = 0.75$ (blue shaded area), and AR(1) processes with parameters estimated from the time series in (a) (red, shaded area). Dashed (dotted) lines mark the ensemble means of the fGn (AR(1) process) respectively.





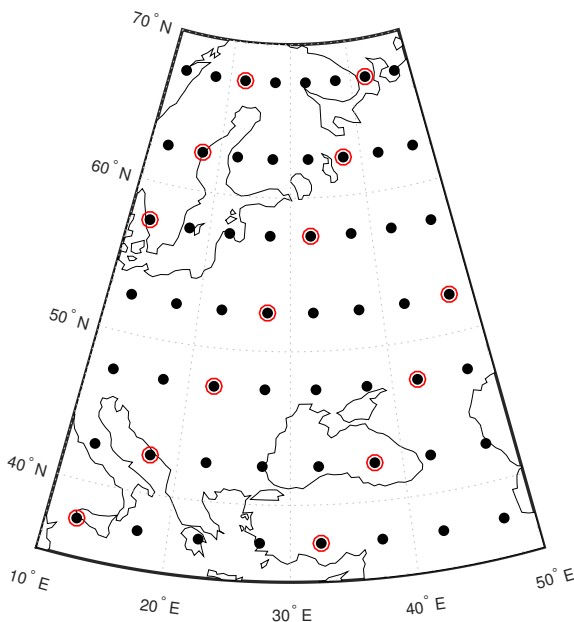

**Figure 2.** The spatial domain of the reconstruction experiments. Dots mark locations of instrumental sites, proxy sites are highlighted by red circles. The superimposed map of Europe provides a spatial scale.





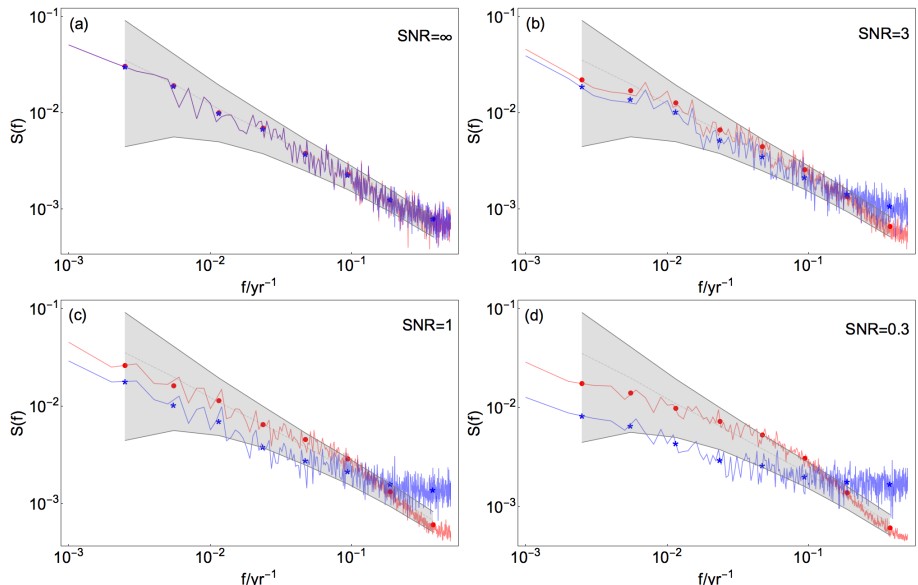

**Figure 3.** Mean raw and log-binned PSD for pseudoproxy data (blue curve and asterisks, respectively) and reconstruction at the same site (red curve and dots, respectively) generated from $\beta_{\text{target}} = 0.75$ and different SNR. Colored gray shadings and dashed, gray lines indicate 95% confidence range and the ensemble mean, respectively, for a Monte Carlo ensemble of fGn with $\beta = 0.75$.





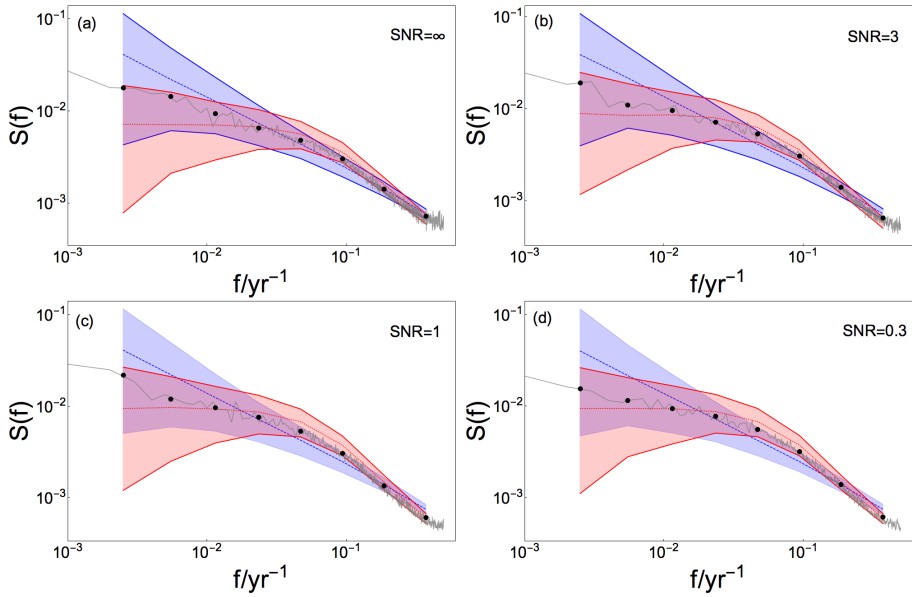

**Figure 4.** Mean raw and log-binned PSD for local reconstructed data at a site between proxies (gray curve and dots, respectively) generated from $\beta_{\mathrm{target}} =0.75$ and different SNR. Colored shadings and dashed/dotted lines indicate 95% confidence range and the ensemble mean, respectively, for a Monte Carlo ensemble of fGn with $\beta =0.75$ (blue) and of AR(1) processes with $\alpha$ estimated from BARCAST (red). The confidence ranges found consistent with the data are drawn with solid lines.





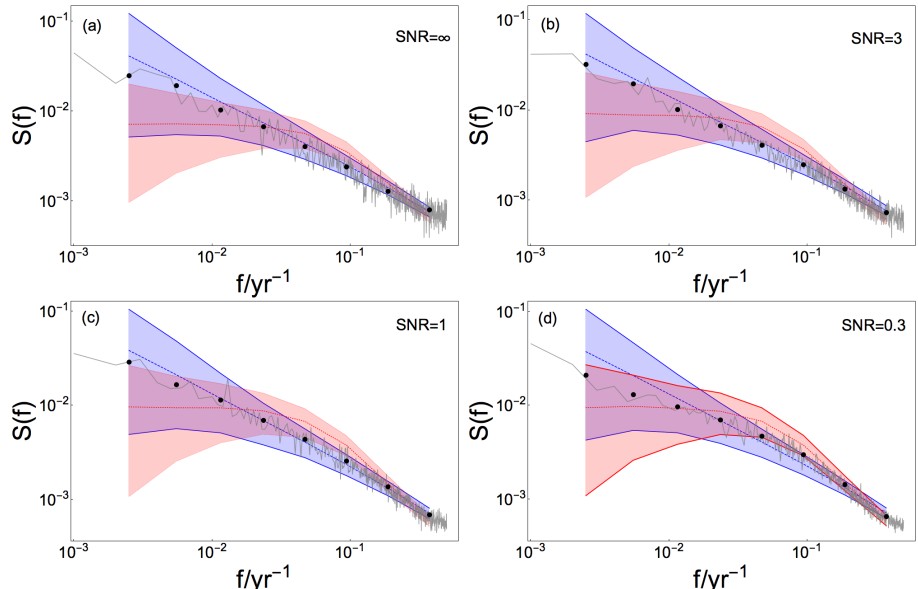

**Figure 5.** Mean raw and log-binned PSD for the spatial mean reconstruction (gray curve and dots, respectively), generated from $\beta_{\text{target}} = 0.75$ and different SNR. Colored shadings and dashed/dotted lines indicate 95% confidence range and the ensemble mean, respectively, for a Monte Carlo ensemble of fGn with $\beta = 0.75$ (blue) and of AR(1) processes with $\alpha$ estimated from BARCAST (red). The confidence ranges found consistent with the data are drawn with solid lines.



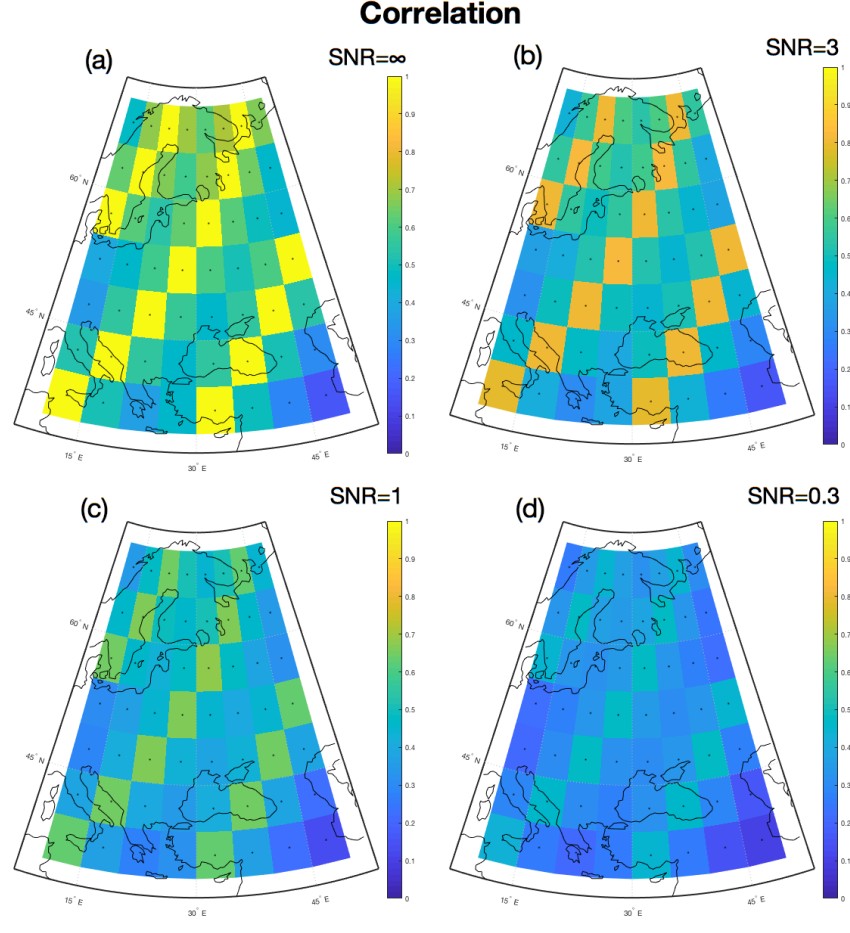

**Figure 6.** Mean local correlation coefficient between reconstructed temperature field and target field for the reconstruction period. $\beta = 0.75$



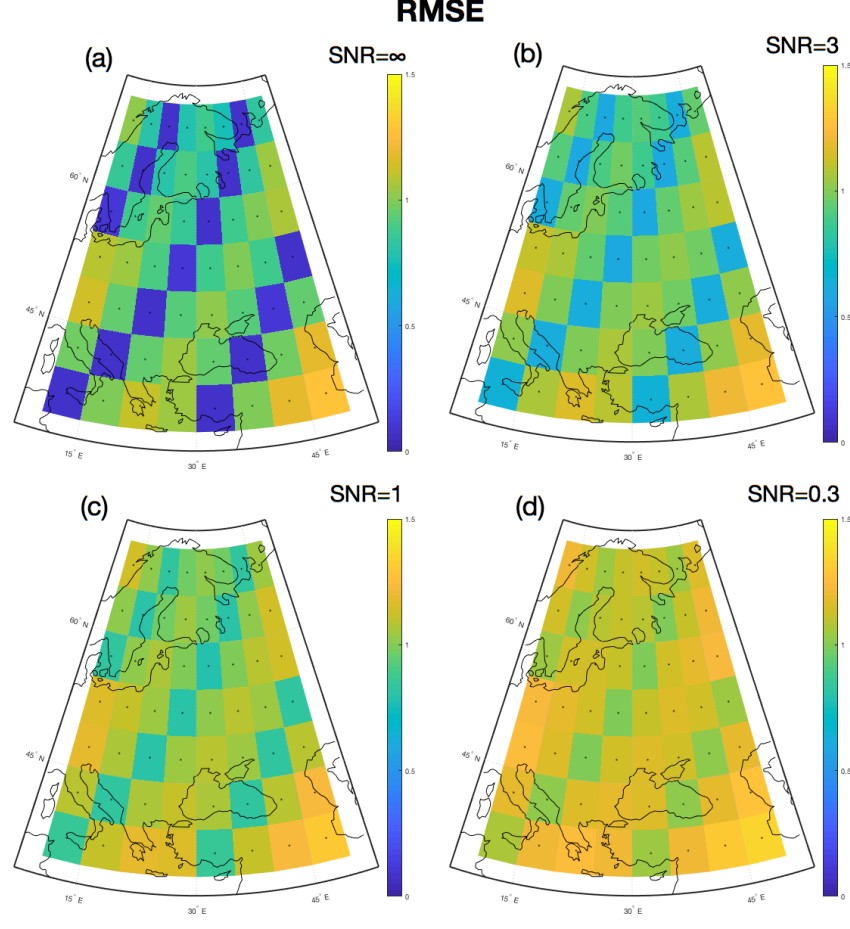

**Figure 7.** Mean local root-mean square error (RMSE) between reconstructed temperature field and target field for the reconstruction period. $\beta =0.75$.





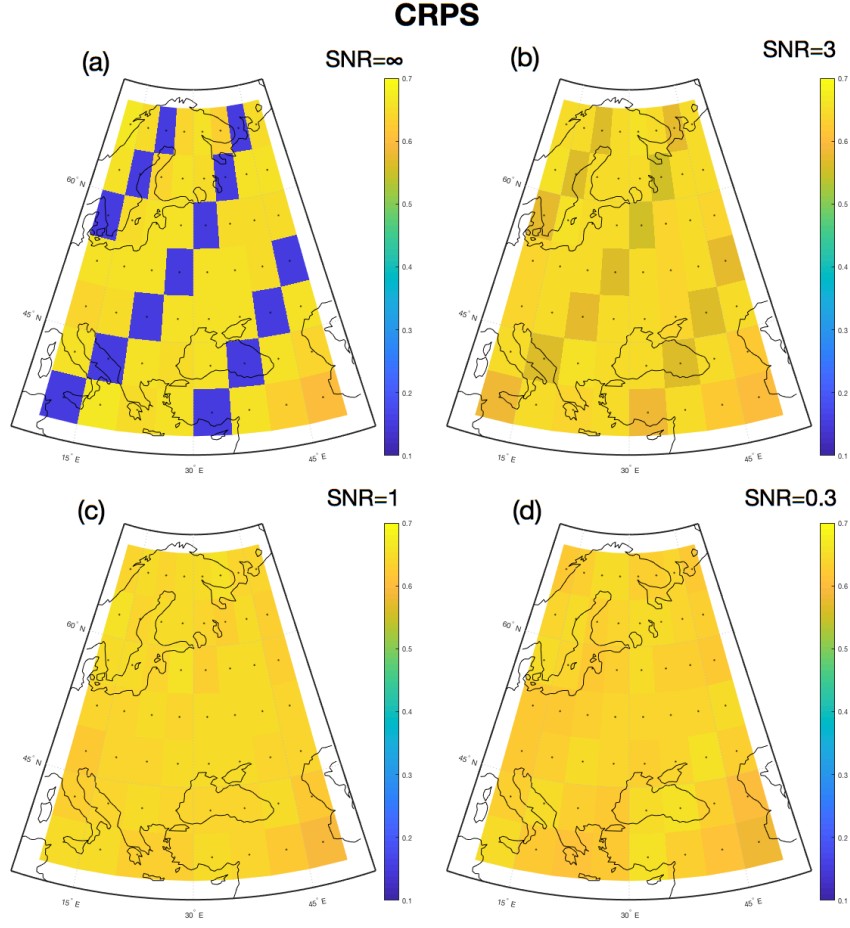

**Figure 8.** Mean local $\overline{\mathrm{CRPS}}_{\mathrm{pot}}$ between reconstructed temperature field and target field. $\beta = 0.75$.





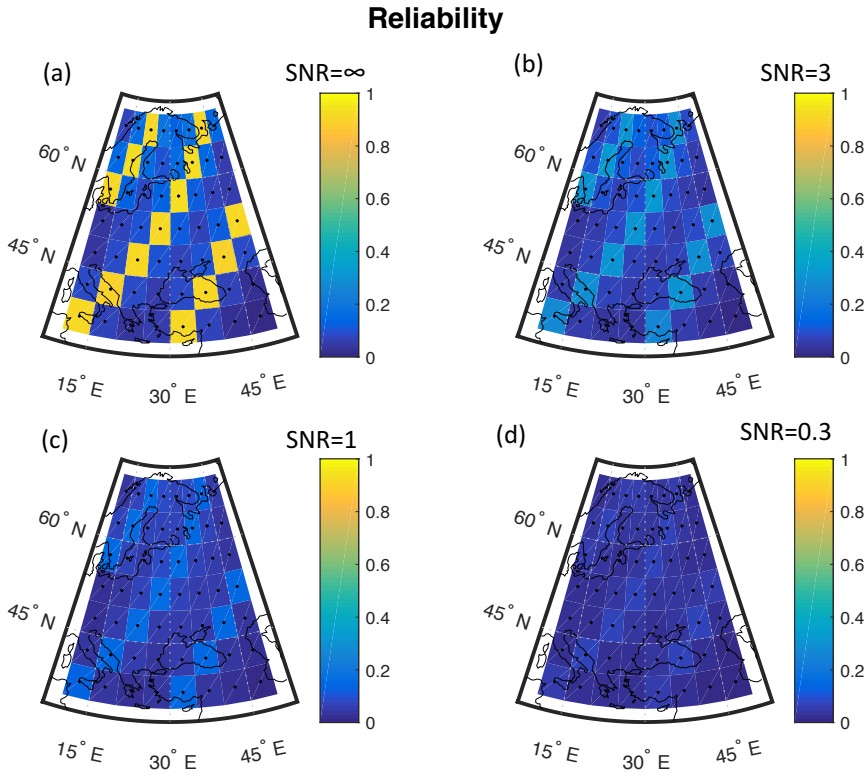

**Figure 9.** Mean local Reliability between reconstructed temperature field and target field. $\beta = 0.75$.




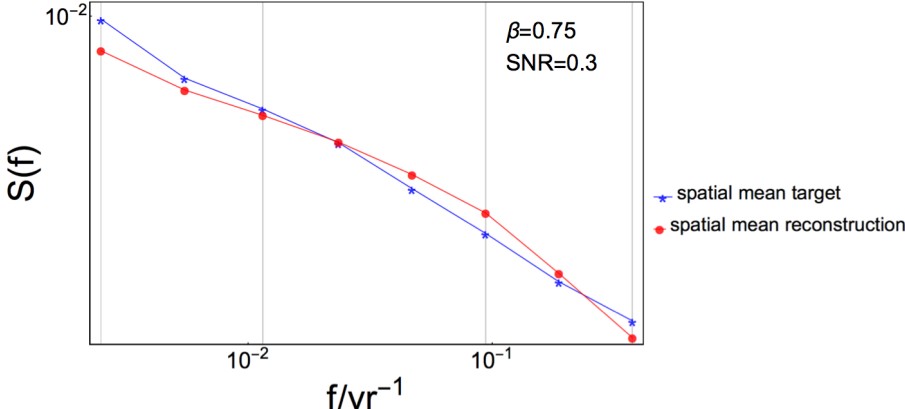

**Figure 10.** Log-log plot showing log-binned power spectra of spatial mean target (blue) and reconstruction (red) for one experiment. Vertical, gray lines mark the frequency ranges used to estimate bias of variance as referred to in Sec. 5.



**Table 1.** Summary of the experiment setup.

| Spatiotemporal resolution: | 5x5 degrees /annual | |
|---|---|---|
| Strength of persistence: | $\beta$=0.55, 0.75, 0.95 | |
| Noise level: | SNR=$\infty$, 3, 1, 0.3 | |
| Iterations before/after thinning: | 5000/500 | |
| | Input data | Reconstruction |
| Ensemble members per experiment | 20 | 30 000 |




**Table 2.** Hypothesis testing results for local reconstructed data compared to Monte Carlo ensembles of fGn and AR(1) processes. The mark
"x" in the table indicates that the null hypothesis cannot be rejected. The null hypotheses 1 and 2 are:

1: The reconstruction is consistent with the fGn structure in the target data for all frequencies.

2: The reconstruction is consistent with the AR(1) assumption from BARCAST for all frequencies.

| Local field values | | | |
|---|---|---|---|
| SNR | $\infty$ | 3 | 1 | 0.3 |
| $\beta = 0.55$ Proxy site | | | |
| 1 | x | x | x | x |
| 2: | | x | x | x |
| $\beta = 0.55$ Between proxy sites | | | |
| 1: | x | x | x | x |
| 2: | x | x | x | x |
| $\beta = 0.75$ Proxy site | | | |
| 1: | x | x | x | |
| 2: | | | | x |
| $\beta = 0.75$ Between proxy sites | | | |
| 1: | x | x | | |
| 2: | x | x | x | x |
| $\beta = 0.95$ Proxy site | | | |
| 1: | x | x | x | |
| 2: | | | | x |
| $\beta = 0.95$ Between proxy sites | | | |
| 1: | x | | | |
| 2: | x | x | x | x |



**Table 3.** Hypothesis testing results for spatial mean reconstructed data compared to Monte Carlo ensembles of fGn and AR(1) processes.
The null hypotheses 1 and 2 are the same as in Table 2, and the "x" has the same meaning.

| Spatial mean values | | | |
|---|---|---|---|
| SNR $\infty$ | 3 | 1 | 0.3 |
| $\beta = 0.55$ | | | |
| 1: x | x | x | x |
| 2: | | | x |
| $\beta = 0.75$ | | | |
| 1: x | x | x | x |
| 2: | | | x |
| $\beta = 0.95$ | | | |
| 1: x | x | x | x |
| 2: | | | |



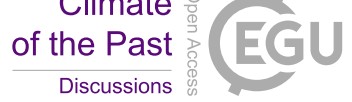

**Table 4.** Mean local skill measures

| SNR | $r$ | RMSE | $\overline{\mathrm{CRPS}}_{\mathrm{pot}}$ | Reliability |
|---|---|---|---|---|
| | $\beta = 0.55$ | | | |
| $\infty$ | 0.65 | 0.74 | 0.53 | 0.30 |
| 3 | 0.54 | 0.93 | 0.63 | 0.12 |
| 1 | 0.43 | 1.04 | 0.64 | $6.6 * 10^{-2}$ |
| 0.3 | 0.29 | 1.16 | 0.62 | $3.1 * 10^{-2}$ |
| | $\beta = 0.75$ | | | |
| $\infty$ | 0.65 | 0.74 | 0.53 | 0.3 |
| 3 | 0.55 | 0.92 | 0.63 | 0.13 |
| 1 | 0.46 | 1.02 | 0.64 | $7.6 * 10^{-2}$ |
| 0.3 | 0.33 | 1.13 | 0.63 | $3.8 * 10^{-2}$ |
| | $\beta = 0.95$ | | | |
| $\infty$ | 0.66 | 0.74 | 0.53 | 0.30 |
| 3 | 0.56 | 0.91 | 0.63 | 0.14 |
| 1 | 0.49 | 0.99 | 0.65 | $9.2 * 10^{-2}$ |
| 0.3 | 0.39 | 1.09 | 0.64 | $5.1 * 10^{-2}$ |





**Table 5.** Mean skill measures for spatial mean

| SNR | $r$ | RMSE |
|-----|-----|------|
| | $\beta = 0.55$ | |
| $\infty$ | 0.97 | 0.17 |
| 3 | 0.86 | 0.29 |
| 1 | 0.75 | 0.38 |
| 0.3 | 0.554 | 0.52 |
| | $\beta = 0.75$ | |
| $\infty$ | 0.95 | 0.17 |
| 3 | 0.85 | 0.29 |
| 1 | 0.8 | 0.37 |
| 0.3 | 0.64 | 0.51 |
| | $\beta = 0.95$ | |
| $\infty$ | 0.96 | 0.17 |
| 3 | 0.87 | 0.28 |
| 1 | 0.82 | 0.36 |
| 0.3 | 0.72 | 0.45 |



**Table A1.** List of parameters defined in BARCAST, form of prior and hyperparameters

| Parameter | Form | Hyperparameters |
|---|---|---|
| $\alpha$ | Truncated normal | $N_{[0,1]}(\alpha_\mu, \alpha_\sigma)$, $\alpha_\mu = 0.5$, $\alpha_\sigma = 0.1$ |
| $\mu$ | Normal | $N(\mu_\mu, \mu_\sigma)$, $\mu_\mu = -0.4$, $\mu_\sigma = 0.1^2$ |
| $\sigma^2$ | Inv-gamma | shape=0.5, scale=0.5 |
| $\phi$ | Lognormal | $\log\phi \sim N(\phi_\mu, \phi_\sigma)$, $\phi_\mu = -7$, $\phi_\sigma = 0.2$ |
| $\tau_I^2$ | Inv-gamma | shape=0.5, scale=0.5 |
| $\tau_P^2$ | Inv-gamma | shape=0.5, scale=0.5 |
| $\beta_0$ | Normal | $N(\beta_{0,\mu}, \beta_{0,\sigma})$, $\beta_{0,\mu} = 0$, $\beta_{0,\sigma} = 0.04$ |
| $\beta_1$ | Normal | $N(\beta_{1,\mu}, \beta_{1,\sigma})$, $\beta_{1,\mu} = 1.14$, $\beta_{1,\sigma} = 0.04$ |





**Table A2.** List of parameter values defined for the target data set. The four values of $\tau_P^2$ listed are related to the four different signal-to-noise ratios: $\mathrm{SNR} = \frac{1}{\tau_P^2}$. $\epsilon_{\mathrm{mach}}$ is machine epsilon, the smallest number represented by the computer which is greater than zero.

| Parameter | Target value |
|---|---|
| $\mu$ | 0 |
| $\phi$ | 1/1000 |
| $\tau_I^2$ | 0 |
| $\tau_P^2$ | $\epsilon_{\mathrm{mach}}$, 0.333, 1, 3.33 |
| $\beta_0$ | 0 |
| $\beta_1$ | 1 |
| $\beta$ | 0.5, 0.75, 0.95 |



**Table A3.** Mean of posterior distribution for each parameters

| Persistence | $\text{SNR}_{\text{target}}$ | $\text{SNR}_{\text{rec}}$ | $\alpha$ | $\mu$ | $\sigma^2$ | $1/\phi$ | $\tau_I^2$ | $\beta_0$ |
|---|---|---|---|---|---|---|---|---|
| $\beta = 0.55$ | $\infty$ | 117.6 | 0.40 | $-2.2 * 10^{-2}$ | 0.83 | 1020 | $2.3 * 10^{-3}$ | $6.7 * 10^{-4}$ |
| | 3 | 3.05 | 0.43 | $-2.6 * 10^{-2}$ | 0.78 | 1053 | $2.4 * 10^{-2}$ | $-2.7 * 10^{-3}$ |
| | 1 | 1.01 | 0.44 | $-3.3 * 10^{-2}$ | 0.75 | 1064 | $3.0 * 10^{-2}$ | $3.8 * 10^{-3}$ |
| | 0.3 | 0.38 | 0.44 | $-2.7 * 10^{-2}$ | 0.75 | 1053 | $3.3 * 10^{-2}$ | $3.2 * 10^{-3}$ |
| $\beta = 0.75$ | $\infty$ | 115.8 | 0.57 | $-3.5 * 10^{-2}$ | 0.68 | 1020 | $2.3 * 10^{-3}$ | $-8.2 * 10^{-4}$ |
| | 3 | 2.81 | 0.62 | $-4.5 * 10^{-2}$ | 0.61 | 1111 | $2.8 * 10^{-2}$ | $5.1 * 10^{-3}$ |
| | 1 | 0.99 | 0.64 | $-5.6 * 10^{-2}$ | 0.59 | 1136 | $3.3 * 10^{-2}$ | $8.3 * 10^{-3}$ |
| | 0.3 | 0.36 | 0.64 | $-5.1 * 10^{-2}$ | 0.59 | 1136 | $3.4 * 10^{-2}$ | $3.8 * 10^{-3}$ |
| $\beta = 0.95$ | $\infty$ | 112.9 | 0.71 | $-4.8 * 10^{-2}$ | 0.5 | 1020 | $2.4 * 10^{-3}$ | $-5.3 * 10^{-4}$ |
| | 3 | 2.69 | 0.77 | $-8.5 * 10^{-2}$ | 0.44 | 1205 | $2.8 * 10^{-2}$ | $3.0 * 10^{-3}$ |
| | 1 | 0.97 | 0.79 | $-9.7 * 10^{-2}$ | 0.41 | 1235 | $3.1 * 10^{-2}$ | $1.1 * 10^{-2}$ |
| | 0.3 | 0.36 | 0.77 | $-1.0 * 10^{-1}$ | 0.42 | 1190 | $2.9 * 10^{-2}$ | $1.6 * 10^{-2}$ |