# Peer review of "How wrong is the BARCAST climate field reconstruction technique in reconstructing a climate with long-range memory?"

_Climate of the Past, 2018_

## Referee Comment (RC1) · Anonymous Referee #1 · 13 Apr 2018

Thanks for the opportunity to review this interesting paper.

Please see the attached files for the content of the review. The first two files contain pp. 1 and 2 of the General and Specific Comments. The third file has further itemized notes for the Specific Comments.

Please note that "minor revisions" is selected for this article. This can potentially change to "major revisions" depending on how the authors respond to the question raised in the third paragraph of the Specific Comments, copied here.

"The most important alteration that may potentially be required concerns the nature of the hypothesis testing confidence intervals the authors utilize. In section 3.1, the

authors describe the Monte Carlo (MC) estimation of the theoretical confidence ranges they utilize for testing of the results against both fractional Gaussian (fGn) and AR1 null hypotheses. Since the actual tests themselves evaluate the mean power spectra of the ensemble experimental results in relation to these distributions, the question arises as to whether it is more appropriate to use estimated theoretical distributions of these processes directly as the authors do, or rather to use estimated distributions of the means of same-sized ensembles of the theoretical values. It is not within the statistical knowledge of this reviewer to resolve this question, but it is asked of the authors to evaluate whether it is applicable in this context."

Please note that the supplemental file is best used when the Comments bar is opened, to see all the comments, etc. as a sequential listing.

Please also note the supplement to this comment:
https://www.clim-past-discuss.net/cp-2018-17/cp-2018-17-RC1-supplement.pdf

———————————————————

[Figure]

**Referee Comment on:**

**How wrong are climate field reconstruction techniques in reconstructing a climate with long-range memory?** Tine Nilsen, Johannes P. Werner, and Dmitry V. Divine (cp-2018-17)

**General Comments**

Overall, this paper provides an excellent addition to the literature concerning the characteristics of climate field reconstructions (CFRs). Specifically, its evaluation of the BARCAST CFR methodology provides an excellent isolation of how the spectral characteristics of spatial and mean field of reconstructions derived using it might/might not be biased by the temporal and spatial specification of the fundamental BARCAST statistical model. By way of context, it is worth noting that there has been interest concerning how the specification of the fundamental statistical model might affect the characteristics of reconstructions derived using BARCAST and related methods. The kind of well-designed, highly-specific experimental design the authors have implemented in order to clearly isolate fundamental characteristics of the method is a highly useful addition in this field.

**Specific Comments**

The results the authors present appear to be well-developed and without substantial technical issue, with the possibility of one exception mentioned in the third paragraph of this section. As the authors note, it does not appear surprising that BARCAST might tend to retain long-term memory information with better fidelity at the spatial mean scale than at the local scale, since the local disturbance term of the spatial covariance specification will tend to average out. Similarly, it is not surprising that the local reconstructions produced by BARCAST are generally of highest quality where there is co-located predictor information, due to the formal specification of the statistical model that necessarily relies on stochastic infilling based on all the model's estimated parameters for the away-from-predictor locations. In this way BARCAST differs from, as one example, CFR methods that reconstruct (typically) orthogonal components of entire fields directly, although such methods necessarily may introduce their own issues regarding the spatial and spectral fidelity of the reconstructions to the true fields they target. It is of interest to utilize a similar experimental design to that presented here to evaluate these, and other, CFR methods.

The authors appear to apply their evaluation criteria conservatively, notably by generally focusing on the fidelity of the entire temporal range of spectral characteristics for the reconstructions, rather than focusing to a significant degree on evaluation of results for specific temporal ranges. The use of metrics that compare cumulative distribution functions (CDFs) is interesting, and in this context it would be significantly helpful if the authors would describe in greater detail what the "Reliability" metric they derive and utilize indicates, beyond providing a

**Fig. 1.** This file is not a figure, but rather p. 1 of the the overall review text

single reference. This is particularly important because one of the key results indicated by these metrics is that there is good agreement with the true target data at co-located data and predictor locations, but that the associated confidence ranges do not show good agreement. This is a tantalizing indication that needs more description, in particular to understand the nature of the confidence range disagreement, which would be highly useful information.

The most important alteration that may potentially be required concerns the nature of the hypothesis testing confidence intervals the authors utilize. In section 3.1, the authors describe the Monte Carlo (MC) estimation of the theoretical confidence ranges they utilize for testing of the results against both fractional Gaussian (fGn) and AR1 null hypotheses. Since the actual tests themselves evaluate the *mean* power spectra of the ensemble experimental results in relation to these distributions, the question arises as to whether it is more appropriate to use estimated theoretical distributions of these processes directly as the authors do, or rather to use estimated *distributions of the means of same-sized ensembles* of the theoretical values. It is not within the statistical knowledge of this reviewer to resolve this question, but it is asked of the authors to evaluate whether it is applicable in this context.

Assuming that the above question is resolved in terms of retaining the existing hypothesis test structure, the most important request for revision is to add explanatory text, along with a smaller set of corrections and other additions, at a number of places in the article. These are all noted in the accompanying PDF document, using the PDF comment and editing capabilities. This request is made to help clarify and contextualize the descriptions in those places for the broader paleoclimatology science community, given the relatively mathematical nature of the article.

Perhaps the most significant of these is to consider adding thoughts about how BARCAST itself might be improved in the Conclusions section. Are there feasible changes to the fundamental temporal and spatial specifications of BARCAST that this research might suggest to consider going forward?

The reviewer thanks the authors and editors for the opportunity to evaluate this quite interesting article.

**Fig. 2.** This file is not a figure, but rather p. 2 of the the overall review text

---

## Referee Comment (RC2) · Anonymous Referee #2 · 13 Apr 2018

Review of CP-2018-17

Here the authors test how the reconstruction method BARCAST performs for target data that has different spatiotemporal covariance structure than what the method assumes. I think this paper addresses an important topic that hasn't really been addressed in the climate field reconstruction community.

I only have minor, mostly presentation-based comments.

I'm not sure that the title of the paper is as informative and accurate as it might be. The title implies that they are making broad claims about all climate field reconstruction techniques when they only tested one. And is long-range memory really the problem?

Isn't it more accurate to say that the problem is when there is long-range memory that a method assumes isn't there or when the method treats it incorrectly? I would recommend adjusting the title to more accurately reflect the work in the paper.

The discussion of the science presented here is quite thorough. Though there are places where I think the text should be tightened and focused. In particular, the abstract takes a long time to set up the problem and discuss the work that was done. I would recommend cutting down and focusing the text of the abstract and also perhaps Section 5, which essentially contains three separate concluding discussion subsections.

p.1 l.2 Citation needed for unsuitability of the CE and RE metrics

Figure 3 caption doesn't explain what each of the (a)-(b) panels are uniquely showing.

I would recommend highlighting in both the abstract and the conclusions how the authors very nicely were able to test the issue of long-range memory in isolation by constructing the spatial fields statistically rather than through climate models. I think this is important to highlight because it's not usually (or ever yet?) done.

───────────────────────────

---

## Referee Comment (RC3) · Anonymous Referee #3 · 16 Apr 2018

Nilsen et al. provide a new test of a critical assumption in CFR, namely how assumptions inherent to various reconstruction techniques may change our interpretation of climate variability on different time scales. They test two main hypotheses, investigating whether surface temperature data can be estimated via pseudoproxy experiments using a Gaussian function with prescribed scaling parameter, or via an AR(1) process with parameters estimated using BARCAST. The paper demonstrates that variability in climate reconstructions may reflect not only the true variability of the proxy data, but might also contain interference / confounding influences of model-fitting that may be incorrect.

[Figure]

This is a very (refreshingly, thank you!) well-written paper and I enjoyed learning about their work. I have some suggestions for minor revisions that might strengthen the paper, but feel that given a bit more work it will prove an important contribution to Climate of the Past.

Namely, I tend to agree with one of the other reviewers that there is not enough direct comparison between the strength of the BARCAST methodology and the other CFR techniques so as to further test the fallout of the assumptions regarding AR1 vs. fGn data. While the authors do review the other techniques in detail which is informative, a direct comparison on the data they've generated in this manuscript using the other available tools would make this much stronger.

The authors also have not included any discussion of the GraphEM CFR technique despite it's inclusion in Wang et al., 2015 (which they cite) – should this CFR method not also be discussed in terms of relative performance? There are also a number of other citations that I believe should be added to the Discussion which I have listed below.

Finally, for the final paragraph of the Concluding Remarks, I was really left hoping for a more forward - looking statement about the future of this field and what your work contributes towards a broader knowledge of our estimates of past climate variability from proxies using these techniques. I think an effort could be made to solidify your findings and put them in a broader scope at the very end of the paper and put your work in context. How does your work enhance our ability as a field to interpret CFRs of past climate variability? What are the truly broad impacts or your work?

Overall, a very nice manuscript. As a general comment, as a person who is not a true expert in CFR, it would be nice if the authors could make an attempt to help readers who may not be as familiar with these topics with real-world examples or layman's terms where appropriate.

Line-by-Line comments: 1.2 comma splice after techniques, remove 2nd line of Introduction, remove 'a' before 'considerable' 2.8 remove "among other things" and revise to "can occur, among other reasons, due to..." 2.19 hyphenate pseudo-instrumental 2.24 Start sentence with and revise 'Available pseudoproxy studies have to a large extent...' 3.1-10 really nice description here! 3.18 'nongaussianity' I think should be non-gaussianity w/hyphen 4.5-20 Why no mention of GraphEM methodology and comparison with like methods? Citation:

Guillot, D., Rajaratnam, B. and Emile-Geay, J., 2015. Statistical paleoclimate reconstructions via Markov random fields. The Annals of Applied Statistics, 9(1), pp.324-352.

4.25 awkward use of comprises – revise to 'Sect. 3 is comprised of an overview' 4.27 change discuss to discusses

Page 12.3 revise "F refers to ..." 12. 16 Figure 9 doesn't get much description or introduction but you partition it away for some reason. Can you please give more information about why you say, for example, 'except Fig. 9' – what is different about Figure 9 exactly? This comes up again on line 12.23.

Page 15 Lines 1-15 I think you need to beef up your discussion here about millennium-long paleoclimate reconstructions, because there are quite a few more citations whose work should be added to the discussion here, especially those which include model-data comparisons, including:

Ault, T.R., Cole, J.E., Overpeck, J.T., Pederson, G.T., Meko, D.M., 2014. Assessing the risk of persistent drought using climate model simulations and paleoclimate data. J. Climate 27 (20), 7529–7549.

Ault, T.R., Cole, J.E., Overpeck, J.T., Pederson, G.T., George, S.St., Otto-Bliesner, B., Woodhouse, C.A., Deser, C., 2013. The continuum of hydroclimate variabil- ity in western North America during the last millennium. J. Climate 26 (16), 5863–5878.

Dee, S.G., Parsons, L.A., Loope, G.R., Overpeck, J.T., Ault, T.R. and Emile-Geay, J.,

2017. Improved spectral comparisons of paleoclimate models and observations via proxy system modeling: Implications for multi-decadal variability. Earth and Planetary Science Letters, 476, pp.34-46.

Laepple, T., Huybers, P., 2014a. Global and regional variability in marine surface temperatures. Geophys. Res. Lett. 41 (7), 2528–2534. http://dx.doi.org/10.1002/2014GL059345.

Laepple, T., Huybers, P., 2014b. Ocean surface temperature variability: large model-data differences at decadal and longer periods. Proc. Natl. Acad. Sci. 111 (47), 16682–16687.

In Section 5.1, you go into the issues with TRW and your estimates of the PSD. The issues with TRW and de-trending methods, especially how this alters the power spectrum, is discussed in Section 3.2.5. of Dee et al., 2017 as listed above and you should compare your analysis to that paper as needed.

---

## Referee Comment (RC4) · Anonymous Referee #1 · 17 Apr 2018

[THIS COMMENT IS A REVISION OF REVIEW "RC1", POSTED 13 APRIL, 2018. IT PROVIDES MINOR UPDATES TO THE FIRST TWO FILES MENTIONED BELOW, FOR THE GENERAL AND SPECIFIC COMMENTS. THERE IS NO SIGNIFICANT EFFECT ON THE MEANING AND PERSPECTIVE OF THE ORIGINAL REVIEW. The third, supplemental, file and the following text remain unchanged.]

Thanks for the opportunity to review this interesting paper.

Please see the attached files for the content of the review. The first two files contain pp. 1 and 2 of the General and Specific Comments. The third file has further itemized notes for the Specific Comments.

[Figure]

Please note that "minor revisions" is selected for this article. This can potentially change to "major revisions" depending on how the authors respond to the question raised in the third paragraph of the Specific Comments, copied here.

"The most important alteration that may potentially be required concerns the nature of the hypothesis testing confidence intervals the authors utilize. In section 3.1, the authors describe the Monte Carlo (MC) estimation of the theoretical confidence ranges they utilize for testing of the results against both fractional Gaussian (fGn) and AR1 null hypotheses. Since the actual tests themselves evaluate the mean power spectra of the ensemble experimental results in relation to these distributions, the question arises as to whether it is more appropriate to use estimated theoretical distributions of these processes directly as the authors do, or rather to use estimated distributions of the means of same-sized ensembles of the theoretical values. It is not within the statistical knowledge of this reviewer to resolve this question, but it is asked of the authors to evaluate whether it is applicable in this context."

Please note that the supplemental file is best used when the Comments bar is opened, to see all the comments, etc. as a sequential listing.

Please also note the supplement to this comment:
https://www.clim-past-discuss.net/cp-2018-17/cp-2018-17-RC4-supplement.pdf

––––––––––––––––––––––––––––

[Figure]

**Referee Comment on:**

**How wrong are climate field reconstruction techniques in reconstructing a climate with long-range memory?** Tine Nilsen, Johannes P. Werner, and Dmitry V. Divine   (cp-2018-17)

**General Comments**

Overall, this paper provides an excellent addition to the literature concerning the characteristics of climate field reconstructions (CFRs). Specifically, its evaluation of the BARCAST CFR methodology provides an excellent isolation of how the spectral characteristics of spatial and mean field reconstructions derived using it might/might not be biased by the temporal and spatial specification of the fundamental BARCAST statistical model. By way of context, it is worth noting that there has been interest concerning how the specification of the fundamental statistical model might affect the characteristics of reconstructions derived using BARCAST and related methods. The kind of well-designed, highly-specific experimental design the authors have implemented in order to clearly isolate fundamental characteristics of the method is a very useful addition in this field.

**Specific Comments**

The results the authors present appear to be well-developed and without substantial technical issue, with the possibility of one exception mentioned in the third paragraph of this section. As the authors note, it does not appear surprising that BARCAST might tend to retain long-term memory information with better fidelity at the spatial mean scale than at the local scale, since the local disturbance term of the spatial covariance specification will tend to average out. Similarly, it is not surprising that the local reconstructions produced by BARCAST are generally of highest quality where there is co-located predictor information, due to the formal specification of the statistical model that necessarily relies on stochastic infilling based on all the model's estimated parameters for the away-from-predictor locations. In this way BARCAST differs from, as one example, CFR methods that reconstruct (typically) orthogonal components of entire fields directly, although such methods necessarily may introduce their own issues regarding the spatial and spectral fidelity of the reconstructions to the true fields they target. It is of interest to utilize a similar experimental design to that presented here to evaluate these, and other, CFR methods.

The authors appear to apply their evaluation criteria conservatively, notably by generally focusing on the fidelity of the entire temporal range of spectral characteristics for the reconstructions. That the authors do focus with some particularity on the highest frequency range is not a contradiction in this regard, as it is warranted given the characteristics of the simulated predictors they utilize. The use of metrics that compare cumulative distribution functions (CDFs) is interesting, and in this context it would be significantly helpful if the authors

**Fig. 1.** This file is not a figure, but rather p. 1 of the the overall review text

would describe in greater detail what the "Reliability" metric they derive and utilize indicates, beyond providing a reference. This is particularly important because one of the key results indicated by these metrics is that there is good agreement with the true target data at co-located data and predictor locations, but that the associated confidence ranges do not show good agreement. This is a tantalizing indication that needs more description, in particular to understand the nature of the confidence range disagreement, which would be highly useful information.

The most important alteration that may potentially be required concerns the nature of the hypothesis testing confidence intervals the authors utilize. In section 3.1, the authors describe the Monte Carlo (MC) estimation of the theoretical confidence ranges they utilize for testing of the results against both fractional Gaussian (fGn) and AR1 null hypotheses. Since the actual tests themselves evaluate the *mean* power spectra of the ensemble experimental results in relation to these distributions, the question arises as to whether it is more appropriate to use estimated theoretical distributions of these processes directly as the authors do, or rather to use estimated *distributions of the means of same-sized ensembles* of the theoretical values. It is not within the statistical knowledge of this reviewer to resolve this question, but it is asked of the authors to evaluate whether it is applicable in this context.

Assuming that the above question is resolved in terms of retaining the existing hypothesis test structure, the most important request for revision is to add explanatory text, along with a smaller set of corrections and other additions, at a number of places in the article. These are noted in the accompanying supplemental document, using the PDF comment and editing capabilities. This request is made to help clarify and contextualize the descriptions in those places for the broader paleoclimatology science community, given the relatively mathematical nature of the article.
 Perhaps the most significant of these potential additions is to consider including thoughts about how BARCAST itself might be improved in the Conclusions section. Are there feasible changes (both mathematically and numerically) to the fundamental temporal and spatial specifications of BARCAST that this research might suggest to consider going forward?

The reviewer thanks the authors and editors for the opportunity to evaluate this quite interesting article.

**Fig. 2.** This file is not a figure, but rather p. 2 of the the overall review text

**Supplement:**

[revised manuscript text omitted]

---

## Author Comment (AC1) · 22 May 2018

Response to reviewer 2

Thank you for providing general comments on the manuscript. The title and abstract of the revised manuscript will be changed, see the general response posted.

Reviewer: p.1 l.2 Citation needed for unsuitability of the CE and RE metrics. Response: see our reply to reviewer one, who has similar concerns.

Reviewer: Figure 3 caption doesn't explain what each of the (a)-(b) panels are uniquely showing. Response: The figure caption will be revised to make this clear.

[Figure]

Reviewer: I would recommend highlighting in both the abstract and the conclusions how the authors very nicely were able to test the issue of long-range memory in isolation by constructing the spatial fields statistically rather than through climate models. I think this is important to highlight because it's not usually (or ever yet?) done. Response: Thank you for this feedback, we will bring the novelty of the data generation into focus as requested. The equations in Sect.2.2 will be moderately rewritten to avoid confusion and misconceptions. The methodology for data generation is unconventional but not unique, Werner and Tingley (2015) generate pseudo proxy data in a similar way, except the target data are formulated directly according to the BARCAST model equations.

References: J. P. Werner and M. P. Tingley. Technical Note: Probabilistically constraining proxy age-depth models within a Bayesian hierarchical reconstruction model. Climate of the Past, 11(3):533–545, 2015. doi: 10.5194/cp-11-533-2015.

---

## Author Comment (AC2) · 22 May 2018

**Response to reviewer 1**

**Specific comments**

*Reviewer* page 1, title: The title needs to be adjusted to reflect that this study is specific to BARCAST, and does not address CFRs generally.
*Response*: The title of the revised manuscript will be changed, see general response.

*Reviewer* page 1, line 14: Does "Selected" here mean that some of the experiments were found to give reconstructions consistent with the AR(1) model, but others did not? If that is the case it needs to be described as such.
*Response*: The abstract will be moderately rewritten, see general response.

*Reviewer* page 2, line 11: This is not necessarily the case, which is why the word "can" needs to be added in line 10.
Ordinary Least Squares still provides optimal parameter estimation when the predictor variable has error, but the predictor variable is not correlated with the predictand error. (Econometrics, Wonnocott and Wonnocott, 2nd ed., 1979, John Wiley and Sons)
In multiple regression, the combined effect across the several predictors can be both reduction or enhancement of the estimated coefficients. (Applied Regression Including Computing and Graphics, Cook and Weinberg, 1999, John Wiley and Sons)
*Response*: Thank you for this additional information, the sentences will be moderately rewritten.

*Reviewer* page 3, line 10: Add a brief parenthetical expression here to denote to the readers what a Lorentzian power spectrum is.
*Response*: A Lorentzian power spectrum has a steep slope at high frequencies, and is flat at low frequencies.

*Reviewer* page 3, line 17: Add a reference here for this statement.
*Response*: References to Fraedrich and Blender (2003); Fredriksen and Rypdal (2016) are inserted.

*Reviewer* page 3, line 17: Note here why Gaussianity is important in this context. It does not seem to follow from the long-range memory properties theme of the rest of the paragraph.
*Response*: The fractional Gaussian noise follows a Gaussian distribution by definition. In order to model the Earth's surface temperature using the fractional Gaussian noise stochastic process, the temperature data cannot deviate too strongly from a Gaussian distribution. This will be made more clear in the revision.
There exists another, more general class of stochastic processes exhibiting LRM but not following a Gaussian distribution, but it is outside the scope of our paper to consider such processes.

*Reviewer* page 3, line 25: Add a reference here for this statement.
*Response*: References to Franke et al. (2013); Fredriksen and Rypdal (2016); Nilsen et al. (2016) are inserted.

*Reviewer* page 5, line 29: A brief explanation of what conjugate means in this context should be added here.
*Response*: conditionally conjugate priors means that the prior and the posterior distribution has the same parametric form. Added in the revision.

*Reviewer* page 3, line 31: Give a reference here for MCMC, the Gibbs sampler, and the Metropolis step. Would Gelman et al., 2003, be appropriate?
*Response*: Yes, reference to Gelman et al. (2003) has been inserted.

*Reviewer* page 7, Eq. 5: The nature of how eq. 6 is derived from eq. 5 may not be obvious/clear to some readers, and perhaps can be explained in a very brief appendix, or possibly by some additional text here.
*Response*: Eq. 6 is not derived from Eq. 5 per se. The exponential kernel function $\exp^{-(1-\alpha)\mathbf{I}(t-s)}$ is replaced with the power-law kernel $(t-s)^{\beta/2-1}$. This is the necessary operation to obtain the desired LRM properties of the target data, the idea stems from Rypdal (2012); Rypdal and Rypdal (2014).

*Reviewer* page 7, line 7: This statement, that there is no contribution from T(0), seems at odds with what is said in the next sentence, where the solution for t > 0 depends not ONLY on the initial condition, but the entire time history of T(t).
[Emphasis of ONLY added.]
This apparent contradiction should be resolved, so the reader is not confused.
*Response*: Sect.2.2 will be modified for clarity. The nature of the long-memory model is that the temperature at time $t$ depends on all past values $[-\infty, t]$. The original form of Eq. 6 is therefore:

$$\mathbf{T}(t) = \int_{-\infty}^{t} (t-s)^{\beta/2-1}\boldsymbol{\epsilon}_{\mathrm{s}}\mathrm{d}s \tag{1}$$

The contribution before time $t = 0$ is then neglected, this means $\boldsymbol{T}(0)=0$ and we get the equation in the same form as Eq. 6 in the manuscript. We apologize for the confusion. Neglecting the contribution prior to $t = 0$ is a choice that is justified in this case.

*Reviewer* page 7, below line 13: Explain why epsilon(sub s) goes away here.
*Response*: $\epsilon_{\mathbf{s}}$ is a function of $s$ only and not of $t$. For the kernel, on the other hand, we have:

$$\lim_{s \to t} (\mathrm{t}-\mathrm{s})^{\beta/2-1} = \infty$$

Hence, this is the term that needs to be adjusted to avoid the singularity.

*Reviewer* page 7, line 14: It can be unclear here to some readers why G is a function of tau, rather than t.

Please explain how this comes to be, and how tau here relates to the tau variance terms in the Normal distributions defined for the e values in Eq 3.

*Response*: A detail was missing in the text, namely that $\tau = t - s$. This $\tau$ is not related to the BARCAST parameters $\tau_I^2$ and $\tau_P^2$. In the revision we will use $t - s$ instead of $\tau$. **G** must be a function of the time step $t - s$, with the unit step function

$$\Theta(t - s) = \begin{cases} 0, & t - s < 0 \\ 1, & t - s \geq 0 \end{cases}$$

*Reviewer* page 8, line 6: Explain this mathematical description more for the readers for whom this expression will not be meaningful as written.

Again, a brief appendix might be useful.

*Response*: The formulas for estimating the periodogram have been slightly rewritten and moved to an appendix. The alternative formulation is less compact and hopefully more readible:

The periodogram is defined here in terms of the discrete Fourier transform $H_m$ as

$$S(f_m) = \left(\frac{2}{N}\right)|H_m|^2, \quad m = 1, 2, \ldots, N/2$$

For evenly sampled time series $x_1, x_2, \ldots x_N$. The sampling time is an arbitrary time unit, and the frequency is measured in cycles per time unit: $f_m = \frac{m}{N}$. $\Delta f = \frac{1}{N}$ is the frequency resolution and the smallest frequency which can be represented in the spectrum.

*Reviewer* page 9, line 14: Wouldn't it be more appropriate here if the confidence range for the theoretical spectrum is generated from the distribution of analogous MEAN spectra generated by the MC process?

*Response*: The approach we are using is meaningful and appropriate. The simulated fGn Monte Carlo ensemble has, on average, the desired statistical properties of theoretical fGn processes. The hypothesis testing involves finding out if the reconstruction ensemble on average has the same statistical properties. The spectral shape is the prominent feature we investigate. The spectral shape of the full fGn Monte Carlo ensemble is practically identical to that of the mean spectrum of the same ensemble. However, the width of the confidence range would be drastically narrowed if the latter type was used, since the mean fGn spectrum is less noisy than the spectrum of an individual ensemble member. Such a narrow confidence range is impractical for our experiment and most other real-world applications, where the data at hand may follow the general power-law

spectral shape, but are not expected to be strictly identical at every single point.

*Reviewer* page 11, line 26: Explain why it is improper in this context.
*Response*: The reduction of error (RE) and coefficient of efficiency (CE) are not suitable for ensemble-based reconstruction in general (Gneiting and Raftery, 2007). For probabilistic forecasts, scoring rules are used to measure the forecast accuracy, and preferably proper scoring rules. Proper scoring rules are built around the concept of reward systems, encouraging the forecaster to be honest, use the state of knowledge or personal beliefs. Proper in this sense means that the maximum reward (CRPS score=0) is given when the true probability distribution is reported. The RE and CE are improper scoring rules, meaning they measure the accuracy of a forecast, but the maximum score is not necessarily given if the true probability distribution is reported. For climate reconstructions, RE=1 and CE=1 implies a deterministic forecast, the maximum score is obtained when the mean (a point measure) within a probability distribution $P$ is used instead of the predictive distribution $P$ itself.

*Reviewer* page 11, line 30: It would be good to add distribution graphics of the skill metrics across their relevant ensembles.
*Response*: This is a good idea, we suggest to include a boxplot in every panel of Figs. 6-9, similar to Figs. 2-5 in Werner et al. (2013).

*Reviewer* page 12, Eq. 11: Explain what the E(sub F) notation means here.
This is done – to an extent – in context in the third sentence following (starting in line 4), but the nomenclature E(sub F) itself should be fully described.
It is not clear what the E operator does.
*Response*: $\mathbb{E}$ is the expectation value. $\mathbb{E}_F$ denotes the expectation value of the cumulative distribution function. Another form of this equation will be used in the revision.

*Reviewer* page 12, line 12: The explanations for average potential CRPS and Reliability need to be augmented with how these are identified in the terms of eq. 11 itself.
This is not clear, and since these are relatively new metrics vis-a-vis paleoclimate reconstruction, it will be highly useful for them to be more concretely explained in terms of the formal mathematical terms of eq. 11.
*Response*: Given the reviewer's interest in these concepts we have decided to rewrite section 4.4 in the revision and include additional explanatory text in the appendix so that it is clear why the RE and CE are improper, and elaborating on the CRPS. See suggested revision in the general reply.

*Reviewer* page 12, line 29: Help explain for the reader what this combination of good agreement with the target, but not with the confidence confidence range, means.
*Response*: We have discovered an error in the calculation of the CRPS, see our general reply. The above statement is no longer correct and will be removed. The recalculated Reliability is consistently low and very close to zero, the average CRPS is therefore to a large extent dominated by the $\overline{\text{CRPS}}_{\text{pot}}$. Sect. 4.4.1 will be rewritten, and Figure 9

removed from the revision.

For visualization of the average CRPS reconstruction skill, see Fig. 1 and 2 below. Figure 1 shows two examples of local target fGn times series with $\beta = 0.75$, SNR $= \infty$ (black) and the 95% confidence range of the reconstruction ensemble in red, based on 1500 ensemble members. The plot (a) is for a proxy location, while (b) is for a location between proxies. For (a) the confidence range is extremely narrow, as the reconstructions are nearly identical to the target. For (b), the ensemble has a larger spread, a few target values fall outside of the confidence range and are therefore considered outliers.

[Figure]

Figure 1: Example of local target (black curve) and 95% confidence range of reconstruction ensemble (red). Parameter $\beta = 0.75$, SNR$=\infty$. (a) is for a proxy location, while (b) is between proxy locations.

[Figure]

Figure 2: $\overline{\text{CRPS}}_{\text{pot}}$ for the parameters $\beta = 0.75$, SNR$=\infty$.

Figure 2 shows the $\overline{\mathrm{CRPS}}_{\mathrm{pot}}$ for the same parameter setup and all locations. This skill score reflects the average accuracy of the reconstructed forecast, best at proxy locations and decreasing away from proxy sites. The low Reliability is not visualized, but in general this indicates that the predicted confidence ranges are in line with the actual reconstructed ensemble. This means that the target value is generally located in the centre of the confidence range, in contrast to high Reliability occurring if the target to large extent is biased towards the upper/lower end of the confidence range. The combination of a low $\overline{\mathrm{CRPS}}_{\mathrm{pot}}$ and high Reliability is still a possible outcome for real reconstructions, as observed for a few locations of Fig. A1 of Werner et al. (2018).

*Reviewer* page 12, line 33: The most general conclusion is that BARCAST performs much better, except re: reliability, where there are co-localized proxy data. This needs to be clearly mentioned here.
This stands in contrast to some other CFR results, where good skill is obtained in regions not co-localized with the predictor data.
E.g., cf. reconstruction of precipitation in California (Wahl et al., 2017, J Clim, DOI: 10.1175/JCLI-D-16-0423.1.)
*Response*: We thoroughly inspected Wahl et al. (2017) which the reviewer is referring to and two more publications (together with supplementary materials) that could be relevant in the context of this comment, namely Wahl and Smerdon (2012) and Diaz and Wahl (2015). Unfortunately, we found no indication that the issue raised by the reviewer has been discussed directly in any of the aforementioned studies. One can assume the reviewer refers to Fig S2 and S4 from Wahl et al. (2017), and the corresponding tree ring proxy network for the region depicted in Fig S2 in Wahl and Smerdon (2012) From these figures, we may infer a somewhat lower, yet mostly non-negative, reconstruction skill for the area centered at ca. 36N 117.5W, featuring a relatively high proxy data density. At the same time, higher skill is obtained in regions without proxy data coverage. However, there is no discussion on the source of these spatial discrepancies (at least we could not find one) other than that in the spatial mean sense, the method demonstrated good performance, which indeed is the case. We take the liberty to speculate that the discrepancies in the reconstruction spatial skill can be caused by a number of factors other than the apparent effect of the choice made on the climate field reconstruction technique. First, target data processing may cause spatial variability in the reconstruction skill. The choice of regridding/interpolation/extrapolation method will have an effect on the target data variance across timescales, especially in the data-poor regions. The second point is that the area with lower RE/CE is associated with proxies from mountainous regions, where regridding might have an effect on the target data sets since climate divides are present.
Should the reviewer find the question critical to be elaborated further, we kindly ask to clarify where this problem is discussed. However, in our opinion this would be fairly difficult to resolve without dedicated experiments with truncated EOF-principal components spatial regression (TEOF-PCSR) on the equivalent target/pseudoproxy datasets.

*Reviewer* page 13, line 4: Indicate which figures and/or tables provide the information

being described as the Discussion section proceeds.

*Response*: This will be done in the revision.

*Reviewer* page 13, line 24: This statement needs to be further illuminated for the reader. Aren't the same equations used for the reconstruction at all sites?

*Response*: These sentences will be rewritten, sorry for the confusion. Yes, the same equations are used for all grid cells, what was meant is that there is already information from the data at proxy sites, overwhelming the priors.

*Reviewer* page 13, line 26: Explain this statement further, in terms of how the resulting reconstruction is influenced.

*Response*: The three levels of BARCAST connects the model equations with the proxy and instrumental data, and with the priors for all parameters and the temperature field. The parameters are therefore interdependent, so adjustment of one parameter must be compensated by the other parameters. The effect of interdependence between the AR(1) coefficient and $\sigma^2$ is demonstrated in Fig. 9 in Tingley and Huybers (2010), where the posterior draws of $\phi$ and $\sigma^2$ are negatively correlated. Tingley and Huybers (2010) claim that for their version of barcast the pdfs of the final draws are not ill-defined, they rather converge to some specific values with relatively narrow pdfs.

For the reconstructions, increasing the AR(1) parameter $\alpha$ means changing the temporal correlation structure of the reconstruction at all frequencies, where the high-frequency component is forced to have stronger temporal correlations and the low-frequency component is forced to have weak or no temporal correlations. Increasing $\beta_0$ means the proxy bias is increased, so the relationship between the proxy and the true temperature is influenced. At last, decreasing $\sigma^2$ means that the white-noise innovations are given less weight, hence the coherent signal is less influenced by local, stochastic perturbation.

*Reviewer* page 13, line 28: This result is not very apparent in Fig. 5 a-c.

*Response*: The sentence mentions specifically "noisy input data", so b-d in all figures. But the effect is indeed minimal in Fig. 5b-c, this will be changed in the revision.

*Reviewer* page 14, line 4: Explain why the characteristic of subsequent years being independent is involved in the discussion here.

That is, relate this characteristic to an incorrect statistical model.

*Response*: The end of this paragraph will be rewritten in the revision:

The assumption of temporal independence corresponds to yet another incorrect statistical model for our target data; a white noise process in time. Note that for target variables/data sets consistent with a white noise process, these types of reconstruction methods are appropriate, as demonstrated using the truncated EOF-principal components spatial regression (TEOF-PCSR) methodology on precipitation data in Wahl et al. (2017).

*Reviewer* page 14, line 23: Explain here the nature of the incorrectness in the confidence intervals.

*Response*: see reply above to comment on page 12, line 29.

*Reviewer* page 15, line 15: Are there thoughts that come from these experiments for how BARCAST itself might be improved?

Are there more realistic characterizations of the fundamental mathematical temporal and spatial specifications that could be recommended?

And if so, what would the numerical estimation ramifications also be?

These kinds of discussion components would be good to add, especially since this study is by construction an evaluation of BARCAST's performance.

*Response*: There are two aspects that will be addressed and elaborated in the revised discussion regarding this comment. It is not only BARCAST that could be improved, the availability of well-documented high-quality proxy records also helps the analyst select an appropriate reconstruction method based on the input data. Investigating the temporal correlation structure is an exercise that can be done at different stages of data manipulation, and we stress that the spatiotemporal covariance structure of reconstructions depends on model/method selection. Hopefully this article can draw the attention of scientists working with proxy data sampling and processing, reconstruction methodology as well as those using such reconstructions for statistical modeling. Our study is meant to improve understanding in these communities, create awareness and enhance communication between them. Modellers need to know as much as possible about what artifacts their data are subject to and which reconstructed time scales are considered most reliable. On the other hand, the producers of proxy data need to know which information is needed in order to provide this information if it exists.

**Point 1 - Improvements regarding proxies**
if the proxy network is of high quality and density, and exhibit LRM properties, the BARCAST methodology without modification should be capable of constructing skillful reconstructions with LRM preserved across the region. This is because the data information overwhelms the vague priors. For assessment of real-world proxy quality it is useful to quantify/model the uncertainties and noise affecting the different proxy types and/or specific reconstructions. Forward modelling of proxies are important tools for this task that we endorse, see for example Dee et al. (2017) for a comprehensive study on terrestrial proxy system modeling and the recent paper by Dolman and Laepple (2018) on forward modelling of sediment-based proxies.

**Point 2 - Improvements of BARCAST**
For the BARCAST CFR methodology, what would drastically improve the performance in our experiments would be reformulation of model Eq. 1-2. However, we cannot guarantee that modifications favoring LRM are practically feasible in the context of a Bayesian hierarchical model, due to higher computational demands. Changing the AR(1) model assumption to instead account for LRM would in the best scenario slow the algorithm down substantially, and in the worst scenario it would not converge at all.

Some cut-off time scale would have to be chosen to ensure convergence. Regarding the spatial covariance structure, accounting for teleconnections introduce similar computational challenges. The more general Matérn covariance family form has already been implemented for BARCAST, but was not used in this study. Another problem is the potential temporal instability of teleconnections. The fact that major climate modes might have changed their configuration through time is now considered realistic. Therefore, setting additional a priori constraints on the model may not be considered justified. The use of exponential covariance structure appears to be a conservative choice in such a situation.

The discussion will be rewritten in the revision to address requests also from reviewers 2 and 3.

*Reviewer* page 15, line 28: In Fig. 3c-d the increased power of the simulated proxy data is at sub-decadal frequencies, not at higher-to-bidecadal frequencies.
If the characteristics mentioned are for real proxy data, that needs to be clarified.
*Response*: The characteristics referred to are the spectra in fig 3c-d, not the real proxy data. To be more precise and general, we will rewrite this sentence:
The characteristic flat spectrum at high frequencies, and the increased power on (sub)-decadal frequencies and lower for (Fig 3c) and Fig. 3d respectively can give the impression that the low-frequency power is inflated.

*Reviewer* page 16, line 1: RCS is only one among a set of tree-ring processing techniques, and this should be made clearer here.
*Response*: You are absolutely right. The focus on the RCS is due to the fact that it is a commonly used method which may cause artifacts as those we discuss. For better balance we will also mention other methods in the revision, such as the age band decomposition (ABD) and signal-free processing mentioned in your next comment.

*Reviewer* page 16, line 11: The "signal free processing" method of tree ring standardization should also be discussed here.
To the extent that the tree ring records involved are also sensitive to moisture, then persistence from one year to the next that is related to biological growth responses to soil moisture persistence can also affect these records. This is different from the standardization issue, pre se.
Cf. Bunde et al., 2013, Nature Clim. Change, doi:10.1038/nclimate1830.
*Response*: This is a good point, which we can state this in the revision for clarity. We are aware of the paper by Bunde et al. 2013, and we will consider citation.

*Reviewer* page 17, line 7: It would be good at this closure point to note how the present study sets a useful precedent for utilizing a carefully-designed, experimental structure to isolate specific reconstruction technique properties – akin to controlled laboratory experiments.
This is noted briefly in line 20 of page 16, but it would be good to highlight this point more broadly, since it is the real power and value-added of this paper.

*Response*: The discussion will be rewritten, and this particular comment will be addressed.

*Reviewer* page 22, Figure 1: Make the lines for the ensemble means significantly wider, so they can be seen better by the reader.
*Response*: Thank you for noting this, it will be fixed for the revision.

**References**

S.G. Dee, L.A. Parsons, G.R. Loope, J.T. Overpeck, T.R. Ault, and J. Emile-Geay. Improved spectral comparisons of paleoclimate models and observations via proxy system modeling: Implications for multi-decadal variability. *Earth and Planetary Science Letters*, 476:34 – 46, 2017. doi: https://doi.org/10.1016/j.epsl.2017.07.036.

H. F. Diaz and E. R. Wahl. Recent california water year precipitation deficits: A 440-year perspective. *Journal of Climate*, 28(12):4637–4652, 2015. doi: 10.1175/JCLI-D-14-00774.1.

A. M. Dolman and T. Laepple. Sedproxy: a forward model for sediment archived climate proxies. *Climate of the Past Discussions*, 2018:1–31, 2018. doi: 10.5194/cp-2018-13.

K. Fraedrich and R. Blender. Scaling of Atmosphere and Ocean Temperature Correlations in Observations and Climate Models. *Phys. Rev. Lett.*, 90:108501, 2003. doi: 10.1103/PhysRevLett.90.108501.

J. Franke, D. Frank, C. C. Raible, J. Esper, and S. Bronnimann. Spectral biases in tree-ring climate proxies. *Nature Clim. Change*, 3(4):360–364, 2013. doi: 10.1038/NCLIMATE1816.

H.-B. Fredriksen and K. Rypdal. Spectral Characteristics of Instrumental and Climate Model Surface Temperatures. *Journal of Climate*, 29(4):1253–1268, 2016. doi: 10.1175/JCLI-D-15-0457.1.

A. Gelman, J. Carlin, H. Stern, and D. Rubin. *Bayesian Data Analysis. 2nd ed.* Chapman & Hall, New York, 2003. 668 pp.

T. Gneiting and A. E Raftery. Strictly proper scoring rules, prediction, and estimation. *Journal of the American Statistical Association*, 102(477):359–378, 2007. doi: 10.1198/016214506000001437.

T. Nilsen, K. Rypdal, and H.-B. Fredriksen. Are there multiple scaling regimes in Holocene temperature records? *Earth Sys. Dynam*, 7(2):419–439, 2016. doi: 10.5194/esd-7-419-2016.

K. Rypdal. Global temperature response to radiative forcing: Solar cycle versus volcanic eruptions. *Journal of Geophysical Research: Atmospheres*, 117(D6), 2012. doi: 10. 1029/2011JD017283.

M. Rypdal and K. Rypdal. Long-memory effects in linear-response models of Earth's temperature and implications for future global warming. *J. Climate*, 27(14):5240–5258, 2014. doi: 10.1175/JCLI-D-13-00296.1.

M. P. Tingley and P. Huybers. A bayesian algorithm for reconstructing climate anomalies in space and time. part —: Development and Applications to Paleoclimate Reconstruction problems. *Journal of Climate*, 23(10):2759–2781, 2010. doi: 10.1175/2009JCLI3015.1.

E. R. Wahl and J. E. Smerdon. Comparative performance of paleoclimate field and index reconstructions derived from climate proxies and noiseonly predictors. *Geophysical Research Letters*, 39(6), 2012. doi: 10.1029/2012GL051086.

E.. R. Wahl, H. F. Diaz, R. S. Vose, and W. S. Gross. Multicentury evaluation of recovery from strong precipitation deficits in california. *Journal of Climate*, 30(15): 6053–6063, 2017. doi: 10.1175/JCLI-D-16-0423.1.

J. P. Werner, J. Luterbacher., and J. E. Smerdon. A Pseudoproxy Evaluation of Bayesian Hierarchical Modeling and Canonical Correlation Analysis for Climate Field Reconstructions over Europe*. *J. Climate*, 26(3):851–867, 2013. doi: 10.1175/ JCLI-D-12-00016.1.

J. P. Werner, D. V. Divine, F. Charpentier Ljungqvist, T. Nilsen, and P. Francus. Spatio-temporal variability of Arctic summer temperatures over the past 2 millennia. *Climate of the Past*, 14:527–557, 2018. doi: 10.5194/cp-14-527-2018.

---

## Author Comment (AC3) · 22 May 2018

**Response to reviewer 3**

*Reviewer*: Namely, I tend to agree with one of the other reviewers that there is not enough direct comparison between the strength of the BARCAST methodology and the other CFR techniques so as to further test the fallout of the assumptions regarding AR1 vs. fGn data. While the authors do review the other techniques in detail which is informative, a direct comparison on the data they've generated in this manuscript using the other available tools would make this much stronger.

*Response*: We agree that this type of comparison would be interesting, but we think that the results of our study are important enough to stand alone. The paper brings in a number of relevant aspects that justifies the anticipated length/extent. These will be elaborated and brought better into focus in the revision, and include the novel method of generating target data and the discussion of proper scoring rules/elaboration on the CRPS.

Though we realize undoubtedly that evaluation of skill for other CFR techniques with a similar set of experiments would be highly relevant, this is unfortunately not an option at this point. The title of the manuscript will be revised to reflect that only BARCAST is considered, see also our answers and consideration in the general reply letter.

*Reviewer*:
The authors also have not included any discussion of the GraphEM CFR technique despite it's inclusion in Wang et al., 2015 (which they cite) ? should this CFR method not also be discussed in terms of relative performance? There are also a number of other citations that I believe should be added to the Discussion which I have listed below.

*Response*: the GraphEM method will be mentioned explicitly in the revision together with the other EM methods as follows:

Other reconstruction techniques that may experience similar deficiencies is the regularized expectation-maximization algorithm (RegEM), (Schneider, 2001; Mann et al., 2007), and all related models (CCA, PCA, GraphEM);

The discussion will be rewritten, and the references mentioned (Laepple and Huybers, 2014b,a) are familiar to us. We will consider citing these papers if appropriate. Furthermore, Ault et al. (2013, 2014) consider precipitation/hydroclimate which do not necessarily exhibit similar persistence properties as surface temperature. As an example, from reviewer 1 we were informed about Wahl et al. (2017), where it is demonstrated that reconstructed precipitation and instrumental data follow a white-noise process in time. For these types of data the multivariate regression-based reconstruction methods may be considered appropriate.

*Reviewer*: Finally, for the final paragraph of the Concluding Remarks, I was really left hoping for a more forward - looking statement about the future of this field and what your work contributes towards a broader knowledge of our estimates of past climate variability from proxies using these techniques. I think an effort could be made to solidify

your findings and put them in a broader scope at the very end of the paper and put your work in context. How does your work enhance our ability as a field to interpret CFRs of past climate variability? What are the truly broad impacts or your work?

*Response*: The discussion and conclusions will be revised, these points will be addressed as they are similar to some comments of reviewer 1.

*Reviewer*: Overall, a very nice manuscript. As a general comment, as a person who is not a true expert in CFR, it would be nice if the authors could make an attempt to help readers who may not be as familiar with these topics with real-world examples or layman?s terms where appropriate.

*Response*: Thank you! Reviewer 1 had many specific comments on subjects that will be described in more detail in the revision. In particular we have tried to extend the text in a number of places to make it more easy to comprehend for potential readers not directly involved in specific studies on CFR methods. If there are other particular segments that are unclear we kindly ask the reviewer to point these out.

**Specific comments**

The grammar has been formatted to comply with the detailed comments, below we respond to the questions the reviewer asks.

*Reviewer*: 12.16 Figure 9 doesn't get much description or introduction but you partition it away for some reason. Can you please give more information about why you say, for example, except Fig. 9? what is different about Figure 9 exactly? This comes up again on line 12.23.

*Response*: We have discovered an error in the calculation of the CRPS, see the general reply for details. Figure 9 will therefore be removed in the revision.

*Reviewer*: Page 15 Lines 1-15 I think you need to beef up your discussion here about millennium- long paleoclimate reconstructions, because there are quite a few more citations whose work should be added to the discussion here, especially those which include model- data comparisons, including: (refs)

*Response*: The focus of this paragraph is very limited, we do not discuss millennium-long paleoclimate reconstructions or model-data comparison in general. It deals with a very specific topic of scale breaks in the power spectrum of Earth's surface temperature over a range of time scales (Lovejoy and Schertzer, 2012; Nilsen et al., 2016). Our results in the present paper demonstrate that such spectral scale-breaks may occur due to proxy noise and/or incorrect model selection, hence the breaks are unrelated to the true climate variability. This means the correlation structure of paleoclimate reconstructions may differ from that of instrumental observations of climate, and it is why the concept of universal scaling laws across a wide range of time scales is perhaps less meaningful.

Regarding the references you mention, some of these are familiar to us and others concern precipitation/hydroclimate which is not our climate variable in focus. In the revised discussion we will try to have better balance and express the above point in a more clear way.

*Reviewer*: In Section 5.1, you go into the issues with TRW and your estimates of the PSD. The issues with TRW and de-trending methods, especially how this alters the power spectrum, is discussed in Section 3.2.5. of Dee et al., 2017 as listed above and you should compare your analysis to that paper as needed.

*Response*: Thank you for making us aware of the Dee et al. (2017) paper which is highly relevant, we will make sure to cite it in the revision. See also the reply to reviewer 1.

**References**

T. R. Ault, J. E. Cole, J-T. Overpeck, G. T. Pederson, S. St. George, B-Otto-Bliesner, C-A. Woodhouse, and C. Deser. The continuum of hydroclimate variability in western north america during the last millennium. *Journal of Climate*, 26(16):5863–5878, 2013. doi: 10.1175/JCLI-D-11-00732.1.

T. R. Ault, J. E. Cole, J. T. Overpeck, G. T. Pederson, and D. M. Meko. Assessing the risk of persistent drought using climate model simulations and paleoclimate data. *Journal of Climate*, 27(20):7529–7549, 2014. doi: 10.1175/JCLI-D-12-00282.1.

T. Laepple and P. Huybers. Ocean surface temperature variability: Large model-data differences at decadal and longer periods. *P. Natl. A. Sci.*, 111(47):16682–16687, 2014a. doi: 10.1073/pnas.1412077111.

T. Laepple and P. Huybers. Global and regional variability in marine surface temperatures. *Geophysical Research Letters*, 41(7):2528–2534, 2014b. doi: 10.1002/2014GL059345.

S. Lovejoy and D. Schertzer. *Low Frequency Weather and the Emergence of the Climate*, pages 231–254. 196. American Geophysical Union, 2012. doi: 10.1029/2011GM001087.

T. Nilsen, K. Rypdal, and H.-B. Fredriksen. Are there multiple scaling regimes in holocene temperature records? *Earth Sys. Dynam*, 7(2):419–439, 2016. doi: 10.5194/esd-7-419-2016.

E.. R. Wahl, H. F. Diaz, R. S. Vose, and W. S. Gross. Multicentury evaluation of recovery from strong precipitation deficits in california. *Journal of Climate*, 30(15):6053–6063, 2017. doi: 10.1175/JCLI-D-16-0423.1.

---

## Author Comment (AC4) · 22 May 2018

**1 General comments to all reviewers and the editor**

We are very grateful for the input from the three anonymous reviewers and hope that all major points have been answered satisfactory. Please see the individual responses to reviewers 1, 2 and 3 on matters that are not discussed in this general reply.

In the revised manuscript, Martin Rypdal will be added as an author as he has contributed significantly to Sect. 2.2 on target data generation. Rypdal's contribution has grown since the original submission so that co-authorship is appropriate.

The most important changes requested by all three reviewers involve (1) changing the title, (2) rewriting the abstract and (3) rewrite discussion/conclusions to be more precise and informative. Other relevant modifications we find necessary is to (4) rewrite/clarify Sect. 2.2 on target data generation to avoid confusion and misconceptions, (5) elaborate on the CRPS skill metric in Sect. 4.4, and inserting selected formulas into the appendix. At last, (6) we have discovered an error in the calculation of the average CRPS scores that influences our skill metric results.

(1) The new suggested title of the manuscript is:

**How wrong is the BARCAST climate field rconstruction technique in reconstructing a climate with long-range memory?**

(2) The abstract will be rewritten and shortened in the revision, as requested. Here is a preliminary suggestion:

> The skill of the state-of-the-art climate field reconstruction technique BARCAST ("Bayesian Algorithm for Reconstructing Climate Anomalies in Space and Time") to reconstruct climate with pronounced LRM characteristics is tested. A novel technique for generating the target data has been developed and is used to provide ensembles of long-range memory stochastic processes with a prescribed spatial covariance structure. Based on different parameter setups, hypothesis testing in the spectral domain is used to investigate if the field and spatial mean reconstructions are consistent with either the fractional Gaussian noise (fGn) null hypothesis used for generating the target data, or the autoregressive model of order one (AR1) null hypothesis which is the assumed temperature model for this reconstruction technique. The study reveals that the resulting field and spatial mean reconstructions are consistent with the fGn hypothesis for most of the tested parameter configurations. There are local differences in reconstructed scaling characteristics between individual grid cells, and the agreement with the fGn model is generally better for the spatial mean reconstruction than at individual locations. Some parameter setups were found to give reconstructions consistent with the AR(1) model, while others were not. Our results show that the use of

target data with a different spatiotemporal covariance structure than the BARCAST model assumption can lead to a potentially biased CFR reconstruction and associated confidence intervals, because of the wrong model assumptions.

(3) Given the opportunity to submit a revised manuscript, the discussion and conclusions will be rewritten to address the comments from all three reviewers. Reformulating this section will take more time, but in general the major changes will be:

- References to which figures/tables we refer to will be made clear as the discussion proceeds.

- Text editing to avoid confusion in several paragraphs, ref comments from reviewer 1 and 3 in particular.

- More precise conclusions regarding the skill metric results using the CRPS and the subcomponents $\overline{\text{Reli}}$ and $\overline{\text{CRPS}}_{\text{pot}}$ (see also 6).

- Consideration of other tree-ring processing techniques, including the age-band decomposition and the signal free processing method.

- Relating our results to to the topic of proxy system modelling, ref. Dee et al. (2017); Dolman and Laepple (2018). Also discuss prospective recommendations to improvements of BARCAST.

- Final conclusions will follow the suggestion from reviewer 1 and summarize how the present study sets a useful precedent for utilizing a carefully-designed, experimental structure to isolate specific reconstruction technique properties.

(4) Rewriting Sect.2.2, see suggestion at page 5-6 of this document.

(5) See suggestion at page 7-9 of this document for elaboration on the CRPS, its subcomponents and explaining why the RE/CE metrics are improper for probabilistic reconstruction methods.

(6) When calculating the skill score metrics, the first year should not be included, since it is used to initialize the reconstruction algorithm and it is not identical to the target value. By accident this year was included when calculating the average CRPS in the original manuscript, and being an outlier of a large magnitude relative to the mean, it had a profound negative effect on the skill metrics we used. After recalculations we find some changes to our results. The $\overline{\text{CRPS}}_{\text{pot}}$ is moderately reduced, see Fig. 1 replacing Fig. 8 in the manuscript.

On the other hand, the Reliability (using overline to indicate average from now on): $\overline{\text{Reli}}$ is drastically reduced after removing the initial year, implying a good consistency

[Figure]

Figure 1: $\overline{\mathrm{CRPS}}_{\mathrm{pot}}$ for $\beta = 0.75$

of the theoretical and empirical CIs. The scores are now consistently below 0.1. The $\overline{\mathrm{Reliability}}$ scores are now of such small orders that we don't find it meaningful to discuss them in the revised manuscript. Hence, Sect. 4.4.1 will have to be rewritten, and figure 9 will be removed. However, the general distinction between the $\overline{\mathrm{CRPS}}_{\mathrm{pot}}$ and the $\overline{\mathrm{Reli}}$ will still be included in the revision.

[revised manuscript text omitted]

---

## Author Response (AR2)

08.06.2018

**Letter to the editor - corrections to manuscript cp-2018-17**

On behalf of the authors I would like to thank the editor Jürg Luterbacher for the fast response that we greatly appreciate, and for constructive comments. The title of the manuscript has been changed again, we now suggest:

**Assessing the performance of the BARCAST climate field reconstruction technique for a climate with long-range memory**

We have also included the reference to Zhang et al. (2018) in the first paragraph as suggested by the editor.
Finally, we have checked again the reviewer comments and corresponding changes made. Overall the revision takes most comments into account. A few sentences/paragraphs have been removed completely to create a better flow and focus. Some references were suggested by the reviewers but are not included in the revision. This includes in particular mentioning/discussing scaling properties in hydroclimate/precipitation records and reconstructions. These variables have their own features that make them different from the temperature variable, and we feel that presenting them in this manuscript requires additional description and context. This would interfere with the line of the presentation, and we prefer to keep the focus on the temperature variable since the discussion is already substantial.

Thank you again and best regards on behalf of the authors,

Dr. Tine Nilsen

**References**

Huan Zhang, Johannes P Werner, Elena Garćia-Bustamante, Fidel Gonźalez-Rouco, Sebastian Wagner, Eduardo Zorita, Klaus Fraedrich, Johann H. Jungclaus, Fredrik Charpentier Ljungqvist, Xiuhua Zhu, Elena Xoplaki, Fahu Chen, Jianping Duan, Quansheng Ge, Zhixin Hao, Martin Ivanov, Lea Schneider, Stefanie Talento, Jianglin Wang, Bao Yang, and Jürg Luterbacher. East asian warm season temperature variations over the past two millennia. *Scientific Reports*, 8:7702, 2018. doi: 10.1038/s41598-018-26038-8.